# Neither "lumpers" nor "splitters": A global revision of Flabellinidae s.l. nudibranchs (Gastropoda: Heterobranchia: Nudibranchia)

Irina Ekimova[1]*, Leila Carmona[2,3], Anna L. Mikhlina[1], Darya Grishina[1], Maria V. Stanovova[1], Dimitry M. Schepetov[4], Craig Hoover[5], Jade de Souza-Canal[5], Karen O. Kuznetsov[1], Ángel Valdés[5]

**1** Biological Faculty, Lomonosov Moscow State University, Moscow, Russia, **2** Departamento de Biología, Facultad de Ciencias del Mar y Ambientales, Campus de Excelencia Internacional del Mar (CEI·MAR), Universidad de Cádiz, Puerto Real, Spain, **3** Instituto Universitario de Investigación Marina (INMAR), Campus de Excelencia Internacional del Mar (CEI MAR), Universidad de Cádiz, Puerto Real, Spain, **4** The Steinhardt Museum of Natural History, Israel National Center for Biodiversity Studies, Tel Aviv, Israel, **5** Department of Biological Sciences, California State Polytechnic University Pomona, Pomona, California, United States of America

* irenekimova@gmail.com

## Abstract

Nudibranch taxonomy is a prime example of the conflict between "lumpers" and "splitters": with rampant cryptic diversity, difficult to interpret morphological characters, plastic ecology, and variable levels of COI divergence among groups. As a result, it is extremely challenging to reach a consensus on what constitutes a species (not to mention a genus or a family). In this paper, we tackle a particularly problematic group, the family Flabellinidae s.l. Recent studies have shown that this group is polyphyletic and have suggested reclassifying the 84 described species into 29 genera across 7 families. However, further studies indicated the instability of the phylogenetic relationships between some of these species, genera, and families and suggested the need for further revisions of this group. The primary goal of the present study is a critical revision of the Flabellinidae s.l. diversity, by testing the monophyly of currently accepted genera and re-evaluating morphological synapomorphies for high-rank taxa, with the incorporation of previously overlooked taxa and some from poorly studied regions. We conducted an updated phylogenetic analysis based on four molecular markers. We also performed morphological analyses based on light and scanning electron microscopy. The analyses confirm the monophyly of Coryphellidae, Apataidae, and Paracoryphellidae. At the same time, the monophyly of Flabellinidae s.str., Samlidae, Unidentiidae, and Flabellinopsidae and the genera *Edmundsella* and *Samla* was questioned. Our results show that *Samla rubropurpurata* represents a new genus described herein as *Launsina* gen. nov. We also identify a new phylogenetic clade represented by specimens from the Kurile Islands, which corresponds to the recently established genus *Mgueolia.* This group is represented by at least four distinct species, three of which are described herein. We demonstrate that the

**Data availability statement:** All relevant data are within the manuscript and its Supporting Information files.

**Funding:** The study was conducted under the state assignment of Lomonosov Moscow State University (IE, AM). The study was supported by Russian Science Foundation grant no. 20-74-10012 for collecting samples in Russian waters, their morphological and molecular analysis (IE, AM, DG, MS). The study of the eastern Atlantic and Mediterranean specimens was funded by CEI MAR Foundation through the scientific improvement axis of the CEI MAR 2022 Plan: Research projects for Young Doctors CEI MAR 2022 (CEI-JD-04) (LC); by the research grant CGL2010-17187, funded by the Spanish Ministry of Economy and Competitiveness (includes the early Ministry of Sciences and Innovation) (LC); by Ramón y Cajal Fellowship RYC2024-049765-I financed by MICIU/AEI /10.13039/501100011033 and the FSE+ (LC). Molecular work at Cal Poly Pomona was supported by the teacher-scholar program funded by the Provost Office and the College of Science (JS, AV). Specimens from New Caledonia were collected during the "Our Planet Reviewed" - New Caledonia expeditions (2016–2019), a joint project of MNHN and Conservatoire d'Espaces Naturels (CEN [now Agence Néo-Calédonienne de la Biodiversité ANCB]), funded mainly by the Gouvernement de la Nouvelle-Calédonie, Province Nord, Agence Française de la Biodiversité (AFB) [now Office Français de la Biodiversité, OFB], and the Lounsbery Foundation (AV). The funders had no role in study design, data collection and analysis, decision to publish, or preparation of the manuscript.

**Competing interests:** The authors have declared that no competing interests exist.

facelinid genus *Bajaeolis* and the recently described genus *Kynaria* are members of the family Flabellinopsidae. Finally, we suggest that the family Piseinotecidae likely represents a senior synonym of the family Unidentiidae. Based on our comprehensive integrative analysis, we propose a new taxonomical scheme for Flabellinidae s.l. and identify consistent common traits and synapomorphies for each high-rank group.

## Introduction

One of the most pervasive tensions in traditional systematics and taxonomy is the conflict between "lumpers" and "splitters" [1]. This issue, mainly driven by reliance on ambiguous morphological traits, resulted in both over- or underestimations of diversity. The existence of cryptic diversity further exacerbated the underestimation problem [2,3]. With the advent of molecular systematics, there was hope that this conflict could be addressed more systematically and objectively, but excessive reliance on arbitrary sequence divergence percentages has produced a sequel to the old debate [4]. Species delimitation algorithms, along with integrative taxonomy, have provided modern taxonomists with more powerful tools to delineate species [5]. However, the conflict rages on, and taxonomists fiercely debate how many species there are. This problem also affects higher taxonomic categories, where the lack of clear definitions can make taxonomic decisions on genera and families highly subjective [6,7]. Nudibranchs are a prime example of this issue: with rampant cryptic diversity, difficult to interpret morphological characters, plastic ecology, and variable levels of COI divergence among groups. As a result, it is extremely difficult to reach a consensus on what constitutes a species—not to mention a genus or a family [6,7]. In this paper, we tackle a particularly problematic group, the family Flabellinidae s.l.

Flabellinidae s.l. is a large cosmopolitan group of nudibranchs, with more than 90 described species [8]. The traditional classification of the family Flabellinidae Bergh, 1889 included two 'main' genera. The genus *Coryphella* Gray, 1850 included species with cerata attached directly to the body, and *Flabellina* McMurtrie, 1831 had cerata arranged in groups on more or less pronounced elevations (or peduncles). Several researchers considered these groups as separate families: Coryphellidae Bergh, 1889 and Flabellinidae, e.g., [9–11]. Further studies identified some "minor" genera, usually monotypic, characterized by either a distinctive rhinophoral morphology (*Coryphellina* O'Donoghue, 1929, *Nossis* Bergh 1902 – papillated rhinophores; *Himatina* Thiele, 1931 – perfoliated rhinophores), radular morphology (*Paracoryphella* Miller, 1971 – polyserial radula vs triserial radula of other Flabellinidae), extensions of the notal margins (*Chlamylla* Bergh, 1886), or by a combination of autapomorphic external and internal traits (*Flabellinopsis* MacFarland, 1966, *Tularia* Burn, 1966) [12–20]. In subsequent papers, most of these "minor" genera were either synonymized with *Coryphella* or *Flabellina* [18,19,21] or were simply ignored due to the unavailability of material for study (e.g., *Chlamylla*, *Tularia*) [19], see also review in [22]. The separate status of the genera *Coryphella* and *Flabellina* was also questioned, precisely because of the presence of substantial morphological diversity

and the existence of transitional forms between them. In most cases, these 'transitional forms' were representatives of the above-mentioned minor genera [23,24]. Cladistic studies confirmed this point of view – the genera *Coryphella* and *Flabellina* were recovered as paraphyletic and thus they were synonymized under the name *Flabellina* [19,23,24]. This 'lumping' taxonomical scheme was dominant for decades, although some researchers did not support the synonymization of *Coryphella* and *Flabellina*, and continued to use the generic name *Coryphella* as valid, e.g., [25–27].

More recently, molecular phylogenetic analyses [22,28] have provided a more nuanced and complex view of the systematics of Flabellinidae s.l. and shown that Flabellinidae diversity is significantly underestimated. The first molecular revision of the group [28] – published online in 2016 – revealed the monophyly of most *Flabellina sensu* Bergh, 1889 diversity (i.e., all species possessing cerata arranged on peduncles – *Flabellina affinis* (Gmelin, 1791) and closely related forms, such as *Flabellina ischitana* Hirano & Thompson, 1990). Species with cerata attached directly to the body (*F. pedata* (Montagu, 1816), *F. verrucosa* (Sars, 1829), *F. nobilis* (Verrill, 1880) and others) were also recovered in a separate and supported clade. Furfaro et al. [28] suggested that this supports restoring the traditional families Flabellinidae and Coryphellidae *sensu* [12].

The comprehensive molecular revision by [22] showed the polyphyly of Flabellinidae s.l. in its traditional composition. Based on molecular phylogenetic analyses using 4 molecular markers, the authors proposed a new taxonomic scheme, identifying 29 genera in 7 families with a total diversity of 84 described species. The authors restored the genera *Coryphella, Himatina, Coryphellina, Paracoryphella, Samla* Bergh, 1900, *Flabellinopsis*, and described 16 new genera and 13 new species. Despite the importance of these results and significant progress made towards understanding the biological and genetic diversity of this group by [22], the "splitting" scheme proposed was problematic. Most of these genera were represented by an extremely small number of species (11 monotypic genera out of 29 genera in total) [22]. As a consequence, many of these genera and families (except the families Apataidae, Samlidae and Unidetiidae) were not accepted by some researchers, who pointed out the inadequate sampling across taxa and the lack of morphological synapomorphies and continued to use the traditional name *Flabellina* for most species of traditional Flabellinidae s.l., e.g., [29–31]. Subsequent studies on the taxonomy and phylogeny of Flabellinidae s.l., which also included phylogenetic approaches, showed the instability of the phylogenetic relationships between some species, genera, and/or families [32,33] and indicated the need for further revisions of this family. Furthermore, a recent revision of the family Coryphellidae [34] proposed a new classification system for this group, and united all nine genera established by [22] into a single genus, *Coryphella*. Most recently, a global re-evaluation of the suborder Aeolidacea [35] has proposed a highly differentiated taxonomic framework, further underscoring the ongoing lack of consensus and the dynamic nature of flabellinid systematics.

In summary, the conflicting taxonomical schemes used by different nudibranch researchers and the unresolved position of several taxa (*e.g.,* the genus *Kynaria*) indicate that the taxonomy of Flabellinidae s.l. requires further revision with the help of a larger taxon sampling and comprehensive morphological analyses. The primary goal of the present study is a critical revision of the Flabellinidae s.l. diversity by testing the monophyly of currently accepted genera and evaluating morphological synapomorphies for high-rank taxa, with incorporation of previously overlooked taxa and species from poorly studied regions.

## Materials and methods

### Collection data

For this study, we used specimens from extensive museum collections, deposited at the California State Polytechnic University Invertebrate Collection (CPIC, Pomona, USA), the Lomonosov Moscow State University (White Sea Branch, ZMMU WS, Moscow, Russia), the Museum of National Scientific Centre of Marine Biology (MIMB, Vladivostok, Russia), Museum National d'Histoire Naturelle (MNHN, Paris, France), Museo Nacional de Ciencias Naturales (MNCN, Madrid, Spain). The sampling was largely focused on the tropical Indo-West Pacific, the Eastern Pacific and the Atlantic. Samples were collected during numerous fieldwork expeditions between 2011 and 2024. Voucher numbers and sampling locations for each specimen are given in S1 Table.

## DNA extraction, amplification, and sequencing

Three different protocols were used to obtain partial fragments of two mitochondrial and two nuclear genes: partial mitochondrial cytochrome c oxidase subunit I and 16S rRNA, and nuclear histone *H3* and 28S rRNA. The protocols used at Lomonosov Moscow State University are described in detail in [34,36,37]; those used at California State Polytechnic University are described in [38]; and those at Instituto Universitario de Investigación Marina (INMAR), University of Cadiz, in [39]. Amplification and sequencing primers are listed in S2 Table. All novel sequences were submitted to NCBI GenBank (S1 Table).

## Data processing and phylogenetic reconstruction

Raw reads for each gene were assembled and checked for ambiguities and low-quality data in Geneious R10 (Biomatters, Auckland, New Zealand). Edited sequences were verified for contamination using the BLAST-n algorithm run over the GenBank nr/nt database [40]. For phylogenetic reconstruction, previously published datasets [22,28,34,35,41] were incorporated, with Tritoniidae selected as a distant outgroup along with representatives of all major Aeolidoidea families. Original data and publicly available sequences were aligned with the MUSCLE [42] algorithm in MEGA 7 [43]. Additionally, all protein-coding sequences were translated into amino acids to verify reading frames and check for stop-codons. Saturation was checked by plotting for all specimens including outgroup the total number of pairwise differences (transitions and transversions) against uncorrected p-distances. For the COI and H3 fragments, saturation was further examined separately for the first, second and third codon positions. Indel-rich regions of the 16S and 28S alignments were identified and removed in Gblocks 0.91b [44] with the least stringent settings. The final sequence dataset contained 328 specimens, including 263 Flabellinidae s.l. specimens, and 65 outgroup taxa. Sequences were concatenated in raxmlGUI 2.0 [45]. Phylogenetic reconstructions were conducted for the concatenated multi-gene partitioned datasets. The best-fit nucleotide evolution models were tested in the MEGA 7 toolkit based on the Bayesian Information Criterion (BIC) for each partition. Multi-gene analyses were done by applying evolutionary models separately to partitions representing single markers. The Bayesian phylogenetic analysis (BI) and estimation of posterior probabilities were performed in MrBayes 3.2 [46]. Two independent analyses were carried out; they were initiated with a random starting tree with 20% of trees discarded as burn-in and ran for $2 \times 10^7$ generations. Markov chains were sampled at intervals of 500 generations. The analysis was converged and terminated when the standard deviation of split frequencies reached <0.01. The Maximum likelihood analysis was conducted in raxmlGUI 2.0 [45] with automatically estimated pseudo-replicate number defined by the autoMRE algorithm [47] under the GTRCAT approximation, applied to each partition individually. Final phylogenetic tree images were visualized in FigTree 1.4.0 and finalized in Adobe Illustrator CS 2015. Node support was assessed with posterior probabilities (PP) for BI and bootstrap values for ML and MP analyses. Only nodes supported by BS ≥ 75 and PP ≥ 0.90 were considered significant [48].

Since our goal is to clarify the taxonomy of Flabellinidae s.l. at the genus and family level, species diversity and identity were not the focus, and no species delimitation analyses were performed. We identified the species-level clades as supported when they demonstrate reciprocal monophyly in combination with stable differences observed in nuclear markers. The p-distances between closely related species were calculated for the COI alignment in MEGA 7 [43].

## Morphological studies

All collected specimens were examined for external morphology under a stereomicroscope. The internal morphology of 91 specimens was examined, including the digestive and reproductive systems. The buccal mass was removed and soaked in hypochlorite solution or in 10% NaOH to dissolve connective and muscle tissues. The radula and the jaws were rinsed in distilled water, air-dried, mounted on an aluminium stub, and sputter-coated with gold for visualization under a CamScan S2 (Cambridge), Quattro SEM (Thermo Fisher Scientific, USA), Jeol JCM7000 (Jeol, Japan), and Nova NanoSEM™ scanning electron microscopes (SEM) available at Lomonosov Moscow State University (Moscow, Russia), at the White

Sea Biological Station (Primorsky, Russia), and the SC-ICYT, University of Cadiz (Puerto Real, Spain) accordingly. Features of the jaws were examined by optical stereomicroscopy and SEM. For the study of the reproductive system, specimens were dissected dorsally along the midline and examined under a stereomicroscope.

### Nomenclatural acts

The electronic edition of this article conforms to the requirements of the amended International Code of Zoological Nomenclature, and hence the new names contained herein are available under that Code from the electronic edition of this article. This published work and the nomenclatural acts it contains have been registered in ZooBank, the online registration system for the ICZN. The ZooBank LSIDs (Life Science Identifiers) can be resolved and the associated information viewed through any standard web browser by appending the LSID to the prefix "http://zoobank.org/". The LSID for this publication is: urn:lsid:zoobank.org:pub: D7D7A473-83EB-4ACF-9036–06300DDBEFB2. The electronic edition of this work was published in a journal with an ISSN, and has been archived and is available from the following digital repositories: PubMed Central, LOCKSS.

## Results

### Molecular phylogeny

In this section the previous classification of [22,35] is used for clarity. A total of 465 newly generated sequences were obtained from 130 specimens of Flabellinidae s.l. The final concatenated dataset included 1724 bp (650 bp for COI, 327 bp for H3, 315 bp for 28S and 396 bp for 16S, the latter two partitions were trimmed with Gblocks).

Both BI and ML analyses revealed similar topologies and compatible node support in most cases (Fig 1, S1 Fig). The analyses confirmed the monophyly of Coryphellidae, Apataidae and Paracoryphellidae (the latter only in the BI analysis, see below). At the same time, Flabellinidae s.str., Samlidae, Unidentiidae and Flabellinopsidae (all sensu [22]), the genera *Edmundsella* Korshunova et al., 2017 and *Samla* were not recovered as monophyletic. This was due to the position of *Flabellina bertschi* Gosliner & Kuzirian, 1990*, F. engeli* Marcus & Marcus, 1968*, F. bulbosa* Ortea & Espinosa, 1998*, Samla rubropurpurata* (Gosliner & Willan, 1991)*, S. telja* (Marcus & Marcus, 1967)*, Piseinotecus soussi* Tamsouri et al., 2014, and *Bajaeolis bertschi* Gosliner & Behrens, 1986*.* In this regard, *F. bertschi* from California clustered with *Edmundsella* sp. CPIC0643 from Florida (posterior probability from BI, PP = 1; bootstrap support from ML, BS = 100), and their clade is sister to Atlantic and Mediterranean species *Edmundsella pedata* and *E. albomaculata* (Pola et al., 2014) (PP = 1; BS = 100). Furthermore, the temperate Eastern Pacific species *Samla telja* was recovered as sister to *Luisella babai* (Schmekel, 1972) (PP = 0.92; BS = 71), and they both clustered with *Flabellina engeli* from Bocas del Toro, Panama (PP = 1; BS = 98) and *Flabellina bulbosa* from Cape Verde (PP = 1; BS = 96). This clade was sister to the main *Samla* group, which included the type species, *Samla bicolor* (Kelaart, 1858) (PP = 1; BS = 100). *Samla rubropurpurata* from Vietnam and New Caledonia were split into two distinct clades (PP = 1; BS = 100), which showed a close relationship with the monophyletic Coryphellidae (PP = 1; BS = 96). The Mediterranean species *Piseinotecus soussi* (Piseinotecidae) clustered with the genera *Pacifia* Korshunova et al., 2017 and *Unidentia* Millen & Hermosillo, 2012 (Unidentiidae, PP = 1, BS = 100) and formed a sister clade to *Unidentia* (PP = 1; BS = 68). Finally, *Bajaeolis bertschi* (Facelinidae) unexpectedly grouped in a single clade with *Flabellinopsis iodinea* (Cooper, 1863)*, Baenopsis baetica* (García-Gómez, 1984) and *Kynaria cynara* (Marcus & Marcus, 1967) (PP = 1; BS = 86).

### Coryphellidae

The monophyly of the genus *Coryphella* was highly supported in BI (PP = 1) and moderate in ML (BS = 70). Within this group, the six monophyletic clades were recovered in BI, which are mostly consistent with previously recovered relationships [40], although the support was lower in most cases. The ML tree suggested a similar but unsupported topology (S1 Fig). The only addition was a single specimen from the Kuril Islands (referred here as *Coryphella* sp. 3), which formed a

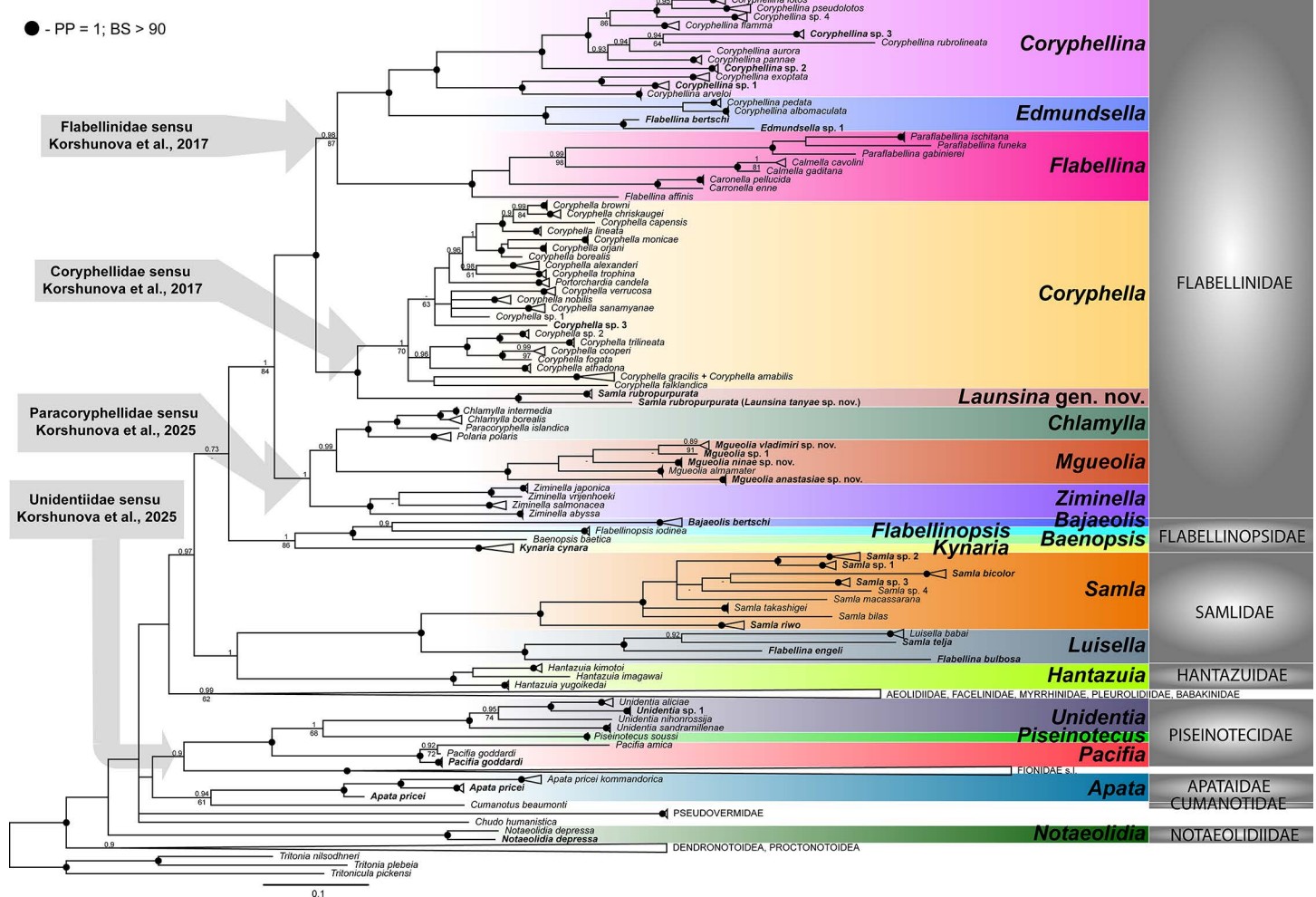

**Fig 1. Molecular phylogenetic hypothesis based on the Bayesian Inference (BI), concatenated dataset of four markers (COI + 16S + H3 + 28S), species-level clades and outgroups are collapsed to a single branch.** Species names on leaves are given according to [22,35], generic and family names on the right indicate revised names. Newly sequenced species are highlighted in bold. Numbers above branches indicate posterior probabilities from BI (> 0.9), numbers below branches – bootstrap support from Maximum Likelihood (ML) (> 60). Nodes supported by maximal support in BI and high support in ML (> 90) are identified with a black circle.

polytomy with most of *Coryphella* diversity. As previously mentioned, *Samla rubropurpurata* was recovered as sister to *Coryphella* (PP = 1; BS = 96).

## Flabellinidae

Within Flabellinidae s.str., three large clades were recovered. The first included all described species of the genus *Coryphellina,* including the type species *Coryphellina rubrolineata* O'Donoghue, 1929*,* and also four putative new species (PP = 1; BS = 99). Within the genus *Coryphellina,* two monophyletic groups were recovered: (1) species with red lines on the dorsal and lateral sides (*Coryphellina rubrolineata* species complex) (PP = 1; BS = 100), and (2) species without lines, i.e., *Coryphellina arveloi* (Ortea & Espinosa, 1998)*, Coryphellina exoptata* (Gosliner & Willan, 1991)*,* and *Coryphellina* sp. 1 (PP = 1; BS = 100).

The second clade was sister to *Coryphellina* (PP = 1; BS = 99) and included the type species of the genus *Edmundsella* (*E. pedata*), its sister species *Edmundsella albomaculata* (PP = 1; BS = 100), and *Flabellina bertschi* and *Edmundsella* sp. clade (PP = 1; BS = 100).

The third clade (PP = 1; BS = 100) contained representatives of the monophyletic genera *Paraflabellina* Korshunova et al., 2017 (PP = 1; BS = 100), *Calmella* Eliot, 1906 (PP = 1; BS = 100), *Carronella* Korshunova et al., 2017 (PP = 1; BS = 100), and the type species of the genus *Flabellina* (*F. affinis*). The genera *Calmella* and *Paraflabellina* were recovered as sister taxa (PP = 0.99; BS = 98), other relationships were not supported.

### Paracoryphellidae

Paracoryphellidae was recovered as a phylogenetically close group to Flabellinidae and Coryphellidae (PP = 1; BS = 84). BI analyses supported its monophyly (PP = 1), but in the ML analysis, this group received much less support (BS = 58). Overall, three major clades were identified. The first included representatives of the genera *Chlamylla* (PP = 1; BS = 99), *Paracoryphella islandica* (Odhner, 1937) (the type species of *Paracoryphella*) and *Polaria polaris* (Volodchenko, 1946) (the type species of *Polaria* Korshunova et al., 2017). The latter was sister to *Chlamylla + Paracoryphella* (PP = 1; BS = 92). The second clade was represented by members of the genus *Ziminella* Korshunova et al., 2017 (PP = 1; BS = 100). Within *Ziminella*, *Z. japonica* (Volodchenko, 1941) was sister to *Z. vrijenhoeki* Valdés et al., 2018 (PP = 1; BS = 100), but the relationships between them, *Z. abyssa* Korshunova et al., 2017, and *Z. salmonacea* (Couthouy, 1838) were not supported. The third clade was highly supported (PP = 1; BS = 100) and included four putative new species from the Kuril Islands, clustering with recently described *Mgueolia almamater* Korshunova, Fletcher & Martynov, 2025 [49]. This group performed a sister relationship to a clade uniting *Chlamylla*, *Paracoryphella* and *Polaria* in BI (PP = 0.99; BS = 31).

### Flabellinopsidae

Flabellinopsidae showed a somewhat unstable position; in the final concatenated analysis, it was recovered as sister to Coryphellidae, Flabellinidae and Paracoryphellidae, but this relationship lacked support in BI (PP = 0.73) and was not recovered in ML. Flabellinopsidae sensu [22] was not recovered as monophyletic due to the position of *Bajaeolis bertschi,* originally described as a member of the family Facelinidae due to its uniserial radula. It formed a monophyletic group with *Flabellinopsis iodinea* and *Baenopsis baetica* (PP = 1; BS = 98). *Kynaria cynara,* which was initially assigned to "Flabellinidae *insertae sedis*" was recovered sister the two flabellinopsid genera and *Bajaeolis* (PP = 1; BS = 86).

### Samlidae

The monotypic genus *Luisella* (the type species *Luisella babai*) formed a single clade with *Samla telja*, *F. engeli*, and *F. bulbosa,* as already mentioned above (PP = 1; BS = 96). With the exclusion of *S. rubropurpurata* and *S. telja,* the genus *Samla* was recovered as monophyletic and highly supported (PP = 1; BS = 100). Specimens initially identified as *Samla bicolor,* were split into five monophyletic groups and further referred as "*Samla bicolor*" species complex.

### Unidentiidae and Piseinotecidae

The monophyly of the family Unidentiidae and both included genera *Unidentia* and *Pacifia*, including *Piseinotecus soussi* was well supported (PP = 1; BS = 100). Within *Unidentia*, one species from Vietnam, likely representing a new species for science, was sister to *U. aliciae* Korshunova et al., 2019 (PP = 1; BS = 95). Within *Pacifia, P. goddardi* (Gosliner, 2010) was not recovered as monophyletic because the specimen CAS182590 formed a polytomy with the other *P. goddardi* specimens and *P. amica* Korshunova et al., 2017 (PP = 1; BS = 100).

## Apataidae

This family was recovered monophyletic with full support (PP = 1; BS = 100). In most trees it clustered with the family Cumanotidae, but the support was not high (PP = 0.94; BS = 61). Within the family, *Apata pricei* (MacFarland, 1966) was split into three clades, each with high support (PP = 1; BS > 95).

## Notaeolidiidae

In both trees, the relationships of *Notaeolidia depressa* Eliot, 1905 with other Flabellinidae s.l. were unresolved. The two specimens displayed high genetic divergence from each other (the p-distance in COI is 9.1%), suggesting they may represent two distinct species.

## Morphological analysis and integration of molecular and morphological data

Morphological traits of all currently accepted Flabellinidae s.l. species (including the families Apataidae, Samlidae, Unidentiidae, the Antarctic family Notaeolidiidae and *Piseinotecus soussi*) are summarized on Fig 2 and in S3 Table, and additionally illustrated on Figs 3–7, S2–S29 Figs. Overall, we identified several features that may vary greatly among species of a single genus, while in other genera, the same features could be identified as a single synapomorphy of each group.

1] Rhinophoral morphology. This feature varies greatly within *Coryphella*, exhibiting mosaic diversity among most other genera or families (Fig 2). However, in the genera *Coryphellina*, *Edmundsella* and the family Unidentiidae, this character is diagnostic: the papillate rhinophores is the most important synapomorphy of *Coryphellina* (S12 Fig), while the smooth rhinophores of *Edmundsella* and Unidentiidae (in addition to other features) differentiate these species from the closely related flabellinid groups. Also, perfoliate rhinophores are characteristic of most Flabellinopsidae (S17A–C, S20A–D, S21A–C Figs), except *Baenopsis baetica* which has rhinophores with numerous ridges (and could be a secondary modification) (S19A Fig) and to most Samlidae, except *Luisella bulbosa* that has rugose rhinophores (S22A–E, S24A, F Figs).

2] Body width. A wide body is apparently a plesiomorphic character, which is typical of Notaeolidiidae, representatives of Paracoryphellidae sensu [22], and several species of the genus *Coryphella* (e.g., *Coryphella nobilis*, *C. trophina* (Bergh, 1890)). A wide body was thought to be a diagnostic character of the family Paracoryphellidae. However, species of the genus *Mgueolia* (Figs 3A, B, 5A–C, 6A, B) may have a slenderer body than *Chlamylla* or *Ziminella* (S10A–C, S16A, D Figs), thus resembling wide-body representatives of the genus *Coryphella* (S6B, C Fig).

3] Reduction of the notal edge. All Flabellinidae s.l. are characterized by different levels of reduction of the notal edge, from the plesiomorphic well-developed and continuous edge of *Chlamylla*, *Ziminella*, *Notaeolidia* (S10A–C, S16A, D, S27A Figs) (in *Mgueolia* it is continuous but less developed than in these groups (Figs 3A, B, 5A–C, 6A, B), to the reduced but continuous of the wide-bodied *Coryphella* (S6B, C Fig), the reduced discontinuous of most *Coryphella* (S6A, D–L Fig) and several members of the *Flabellina* s.l. clade, as well as *Coryphellina*, *Edmundsella*, Flabellinopsidae and *Samla* (see S2, S12, S15A–C, S17A–C, S19A, S20A–D, S21A–C, S22A–E Figs). A completely reduced notal edge could be observed in several members of the *Flabellina* s.l. clade (S3 Table), in *Launsina* gen. nov. (Fig 7A, B) and in Piseinotecidae, Apataidae and *Luisella* (S3 Table, S24A, F, S25A–E, S29A–C Figs). Korshunova et al. [22] reported a completely reduced notal edge in *Coryphella athadona* Bergh, 1875, which is not supported either by our data (S6L Fig), or by the morphological redescription of this species [50].

4] Ceratal position and arrangement. This character was found to be one of the most important, not only for separating the families, but also for generic delineation. Our analysis showed that species of the Samlidae clade (Figs 1 and 2) have a similar arrangement of cerata: this could be interpreted as 'serial', when all cerata are placed on a single simple peduncle, but the length of cerata gradually decreases from the dorsal to the ventral side (Fig 2, S22A–E, S24A, F Figs). Furthermore, representatives of the family Apataidae are characterized by comb-shaped peduncles (Fig 2,

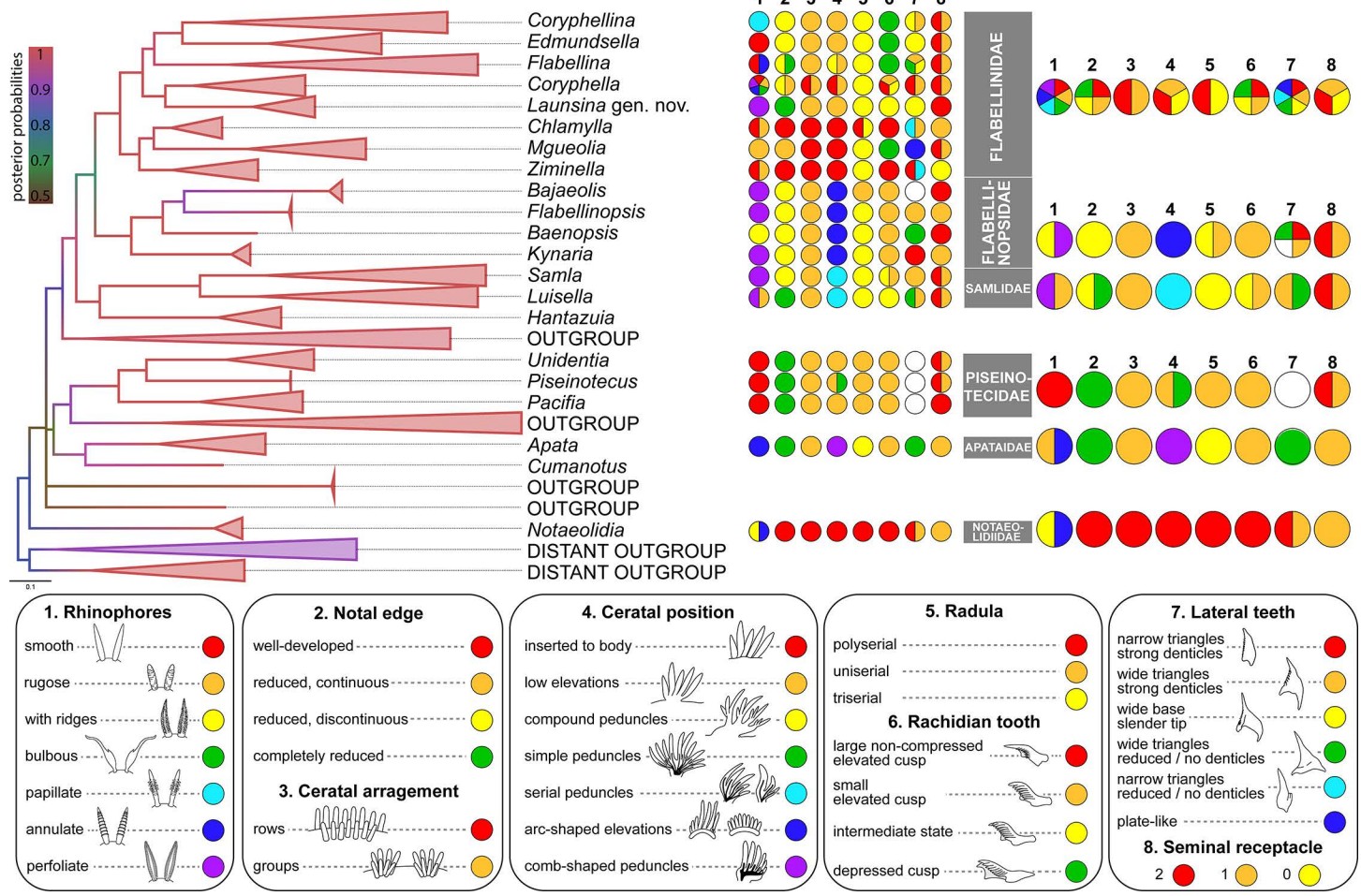

**Fig 2. Summary of diagnostic characters of the flabellinid taxa and the family Notaeolididae, based on the original data and literature search for the morphology of each species represented in S3 Table.** Phylogenetic reconstruction based on Bayesian Inference is given on the left, and each chart pie on the right refers to the character states for the respective genus (center) or family (right). Character states for the family Apataidae are indicated with consideration of the diagnosis of the genus *Tularia.* The legend for diagnostic character states coding each color of circles is given on the bottom.

S25A–E Fig). In Flabellinopsidae all genera have cerata placed on wide arc-shaped elevations (Fig 2, S17A–C, S19A, S20A–D, S21A–C Figs), which are not typical for members of closely related Flabellinidae s.l. Finally, across clades corresponding to Coryphellidae, Flabellinidae, and Paracoryphellidae (all sensu [22]), cerata might be arranged in rows or groups (Fig 2). In the latter case, they are either attached directly to the body (most *Coryphella,* all *Chlamylla, Ziminella* and *Mgueolia* (Fig 3A, B, 5A–C, 6A, B, S6, S10A–C, S16A, D Figs)), or placed on low elevations (all *Coryphellina, Edmundsella, Launsina* gen. nov., few *Coryphella,* few *Flabellina* s.l*., see* Fig 7A, B, S2B, S12, S15A–C Figs), or placed on compound peduncles (most *Flabellina* s.l., see S2A, C–F Fig) (Fig 2, S3 Table).

5] Position of the anal opening. Most flabellinids are distinctly pleuroproctic animals, however, there are several taxa with the anus placed more dorsally, the so-called 'pleuroproctic in a higher acleioproctic position' (Apataidae, most Piseinotecidae) or acleioproctic (some *Flabellina, Unidentia, Piseinotecus soussi*). The more dorsal position of the anal opening evolved independently in different groups. Also, it is worth mentioning that in the majority of the Flabellinidae s.l. the

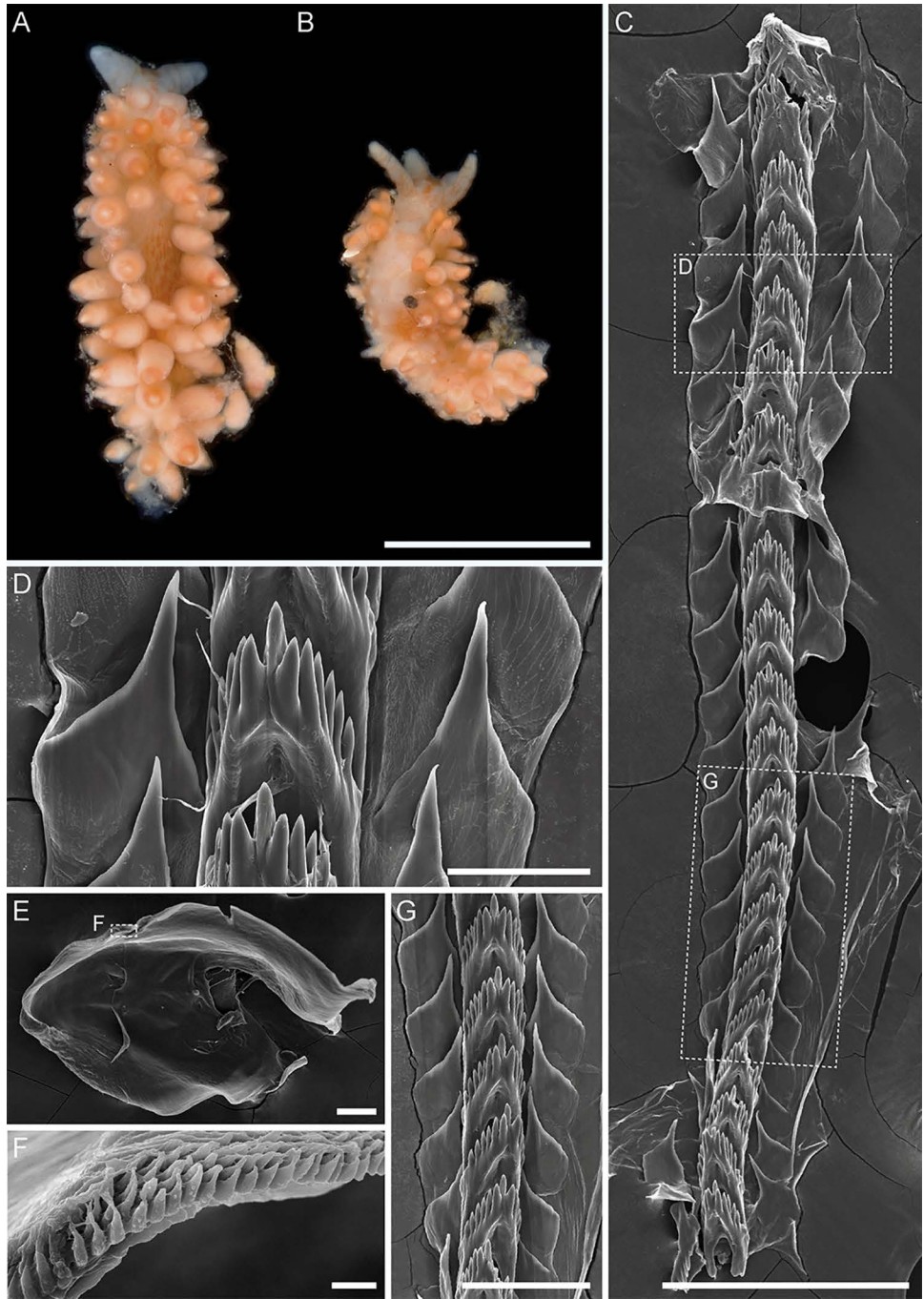

**Fig 3. External and internal morphology of _Mgueolia vladimiri_ sp. nov.** A – paratype MIMB52146, living photo. B – holotype MIMB52148, living photo. C – paratype MIMB52146, radula. D, G – paratype MIMB52146, fragment of radula, enlarged. E – paratype MIMB52146, left jaw plate. F – paratype MIMB52146, denticulation of the masticatory process. Scale bars: A, B = 5 mm; C = 250 μm; D = 50 μm; E, G = 100 μm; F = 10 μm.

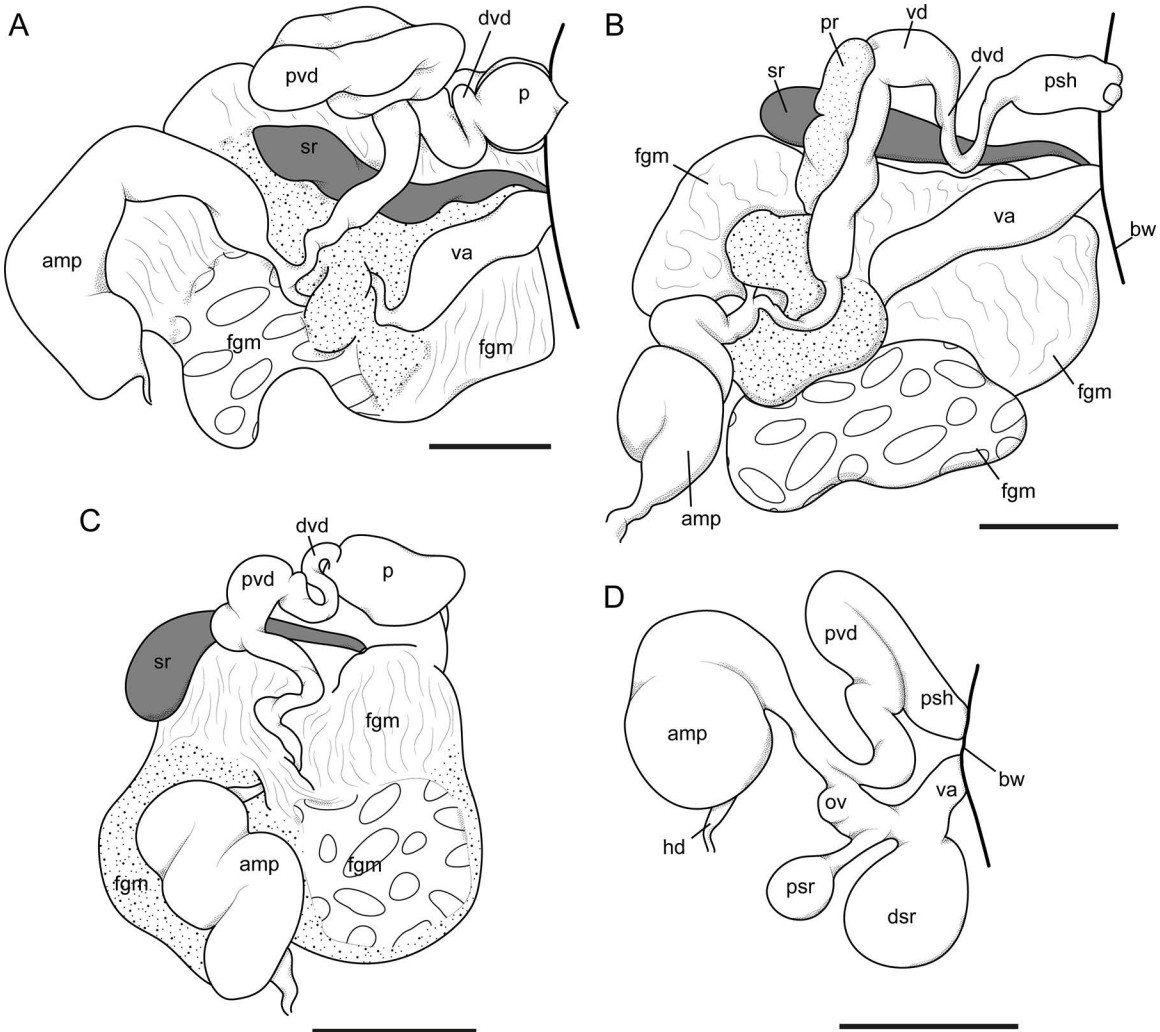

**Fig 4. The reproductive system morphology of representatives of *Mgueolia* and *Launsina* gen. nov.** A – *Mgueolia vladimiri* sp. nov., holotype MIMB52146. B – *Mgueolia anastasiae* sp. nov., holotype MIMB52144. C – *Mgueolia ninae* sp. nov. holotype MIMB52149. D – *Launsina tanyae* sp. nov., holotype MIMB52174, female gland mass removed. Abbreviations: amp – ampulla, bw – body wall; drs – distal seminal receptacle, dvd – distal vas deferens, fgm – female gland mass, hd – hermaphroditic duct, ov – oviduct, p – penis, pr – prostatic region, psh – penial sheath, psr – proximal seminal receptacle, pvd – prostatic vas deferens, sr – seminal receptacle, va – vagina, vd – vas deferens. Scale bars: 500 µm.

anus is placed in a somewhat anterior position, but in the *Mgueolia* and *Chlamylla* clades as well as in the wide-bodied *Coryphella* it is placed closer to the middle of the body, and in *Ziminella* and *Notaeolidia* it is located laterally on the second half of the body. We reviewed the cleioproctic position of the anus in *Bajaeolis bertschi* as mentioned in the initial description of this species [51]. Although the anus is located right under the postcardial arc-shaped elevation with cerata, the reduced notal edge is distinct and observed above the anal opening, thus its position should be identified as pleuroproctic (S3 Table).

6] Radular formula. The triserial condition of the radula is characteristic for the majority of Flabellinidae s.l. (Fig 2, S3 Table). Few exceptions are: the uniserial radula of *Bajaeolis bertschi* (Flabellinopsidae) (S21E–G Fig), and the uniserial radula in Piseinotecidae (including former Unidentiidae) (Figs 1, 2, S29D–F Fig). *Notaeolidia* possesses a polyserial

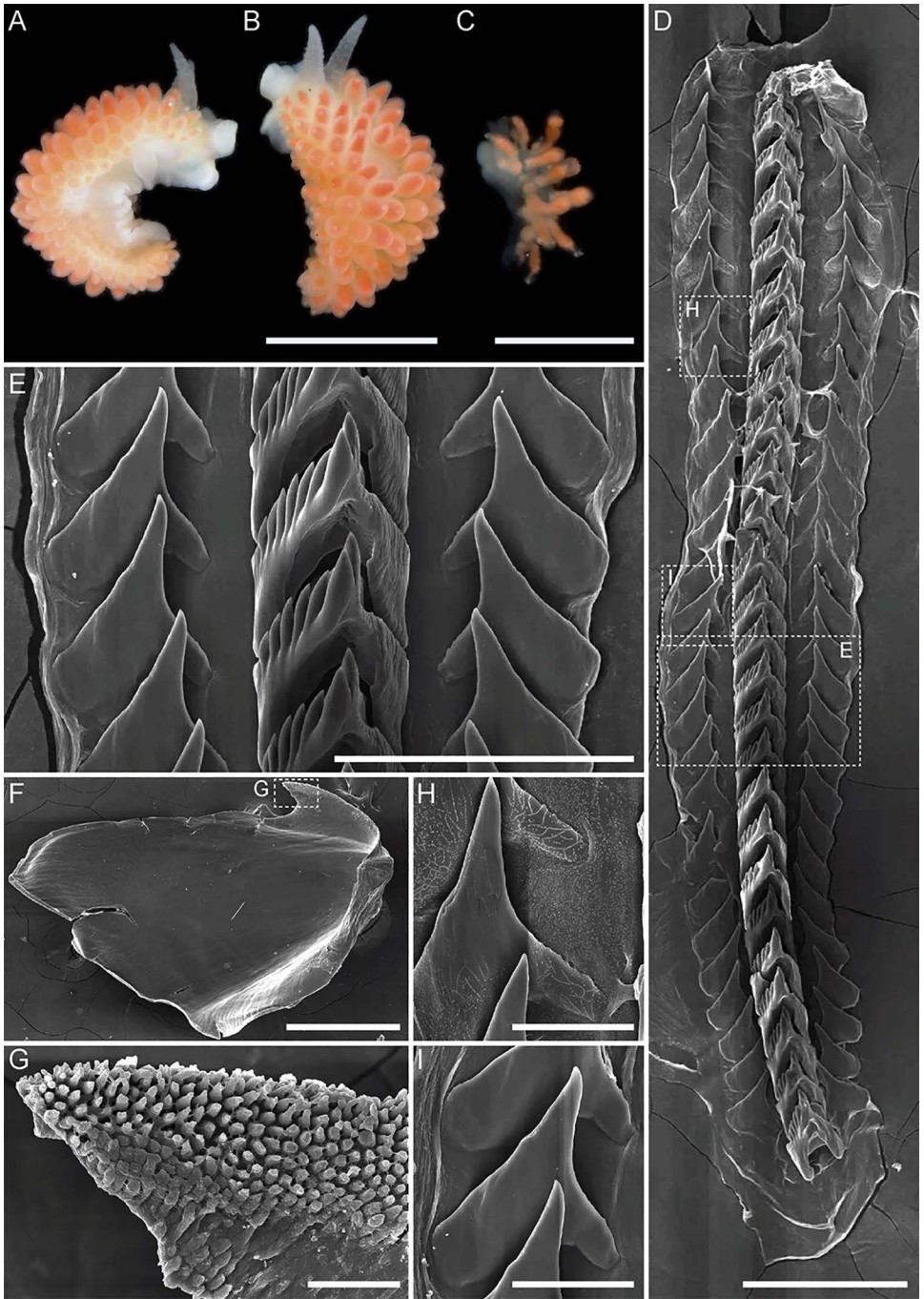

**Fig 5. External and internal morphology of _Mgueolia anastasiae_ sp. nov.** A, B – holotype MIMB52144, lateral (A) and dorsolateral (B) view, living photos. C – paratype MIMB52145, living photo. D – holotype MIMB52144, radula. E – holotype MIMB52144, fragment of radula, enlarged. F – holotype MIMB52144, right jaw plate. G – holotype MIMB52144, denticulation of the masticatory process. H, I – holotype MIMB52144, serrated (H) and smooth (I) lateral teeth. Scale bars: A, B = 5 mm; C = 2 mm; D = 250 µm; E = 150 µm; F = 500 µm; G = 25 µm; H, I = 50 µm.

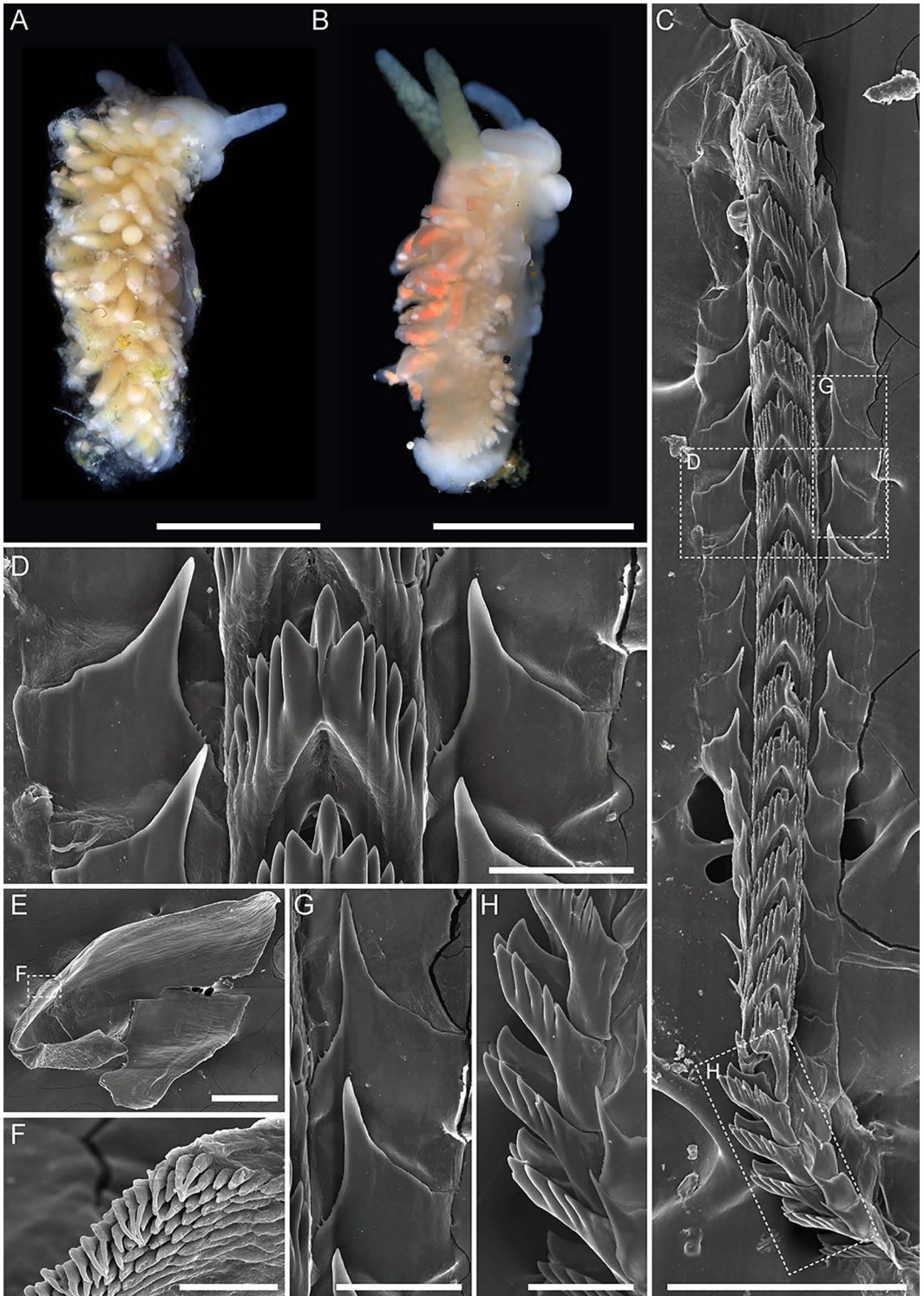

**Fig 6. External and internal morphology of *Mgueolia ninae* sp. nov.** A – holotype MIMB52149, living photo. B – MIMB52209, living photo. C – holotype MIMB52149, radula. D – holotype MIMB52149, fragment of radula, enlarged. E – holotype MIMB52149, left jaw plate. F – holotype MIMB52149, denticulation of masticatory border. G, H – holotype MIMB52149, details of lateral (G) and rachidian (H) teeth. Scale bars: A = 2,5 mm. B = 5 mm. C = 500 µm. D, G, H = 50 µm. E = 250 µm. F = 25 µm.

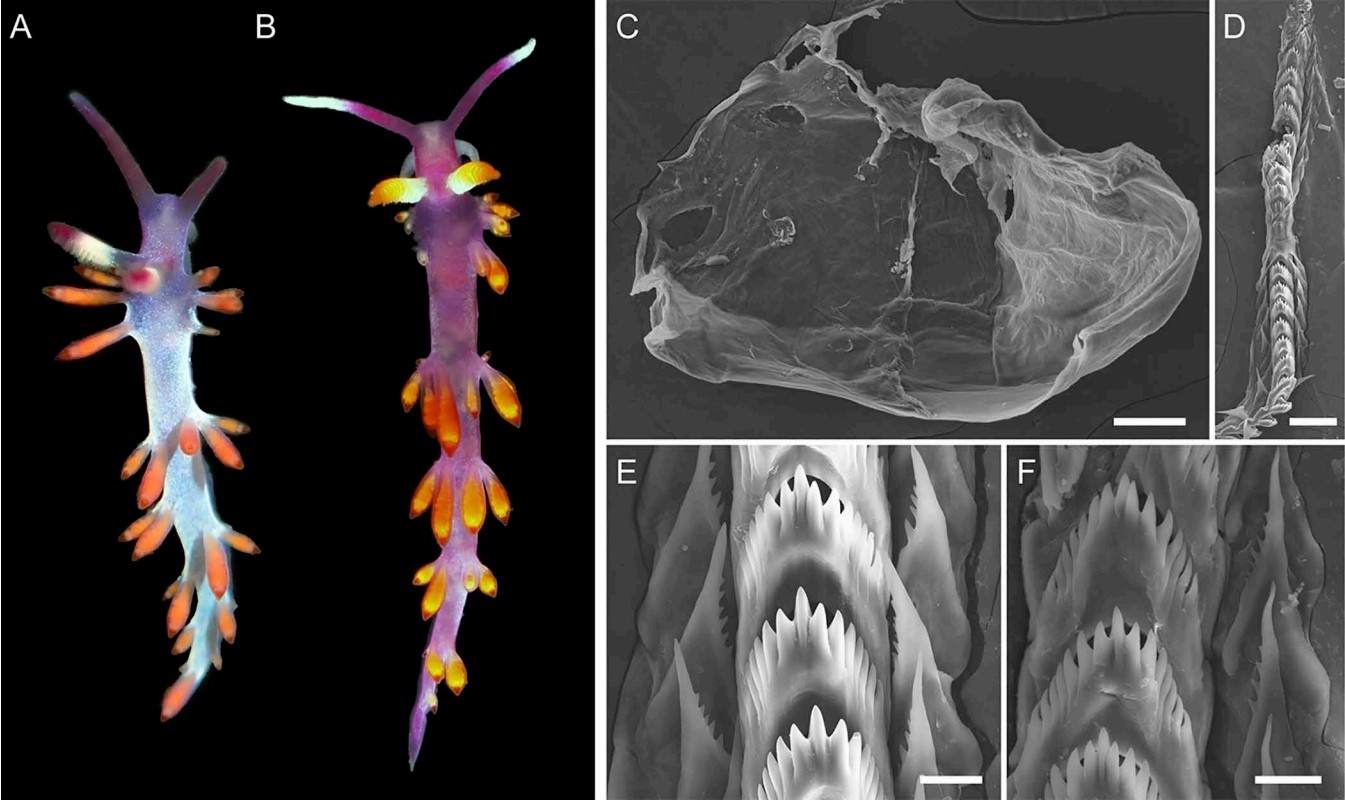

**Fig 7. External and internal morphology of representatives of the genus *Launsina* gen. nov.** A – *Launsina rupropurpurata* comb. nov., MNHN-IM-2019-27166, living specimen, ca. 6 mm in length preserved. B–F – *Launsina tanyae* sp. nov., holotype MIMB52174, 7 mm in length. B – living specimen. C – jaws. D – Radula. E – anterior radular portion. F – posterior radular portion. Scale bars: C, D = 50 μm, E, F = 10 μm.

radula, a unique condition among Aeolidida (Fig 2, S27B, C, F Fig). The same was also observed [18,22] in juveniles of the genus *Paracoryphella* sensu [22], which represents a plesiomorphic trait.

7] Morphology of the central cusp of the rachidian teeth. The position of the central cusp is one of the key differences between the genera of Flabellinidae, Coryphellidae, and Paracoryphellidae as defined by [22]. The depressed central cusp is observed in *Mgueolia, Edmundsella, Flabellina,* and *Coryphellina* (Figs 2, 3D, G, 5E, 6D, H, S3A–E, G, S13A–F, S15E–G Fig)*,* while in *Coryphella* it is somewhat elevated (S7 Fig), even when the cusp is slightly compressed by adjacent denticles (S7B Fig). In most *Coryphella* species, the cusp is longer than the lateral denticles (S7 Fig). The central cusp maybe large and broad, much larger than the lateral denticles, as in *Ziminella* and *Chlamylla* (S10D–F, S16B, E Figs). Overall, the depression/elevation of the central cusp is a more important feature for generic differentiation, while its dimensions and the number and morphology of the lateral denticles are useful features for species-level taxonomy.

8] Lateral teeth shape and denticulation. This is a quite variable feature, but we identified six major patterns (Fig 2): narrow (1) or wide (2) slightly curved triangular teeth with a large central cusp and strong lateral denticles; triangular teeth with a wide base and a slender subulate tip (3); wide (4) or narrow (5) triangular teeth with reduced lateral denticles or denticles absent; and plate-like wide triangular teeth (6). The morphology of the lateral teeth in general correlates

with the recovered phylogenetic relationships (Fig 2), however this feature may vary even within a single clade (e.g., in *Coryphella, Ziminella* two morphological types are observed) (S3 Table), and likely has less phylogenetic significance.

9] Number and position of seminal receptacles. This feature was believed to have a high phylogenetic significance, however our review (S3 Table) in combination with newly obtained molecular phylogenetic hypotheses (Figs 1 and 2) shows that the number, position and morphology of the seminal receptacles vary greatly even within a single genus. For example, species of the genus *Edmundsella* were reported to have one bilobed seminal receptacle (*E. bertschi, E. pedata, E. vansyoci* (Gosliner, 1994)), while in *E. albomaculata* and *E. dushia* (Marcus & Marcus, 1963) both proximal and distal seminal receptacles are present (S14C, D Fig). The same is true for the well-defined genus *Coryphellina,* where usually two receptacles were present, and the proximal receptacle was bilobed (S14A Fig), but in *C. albomarginata* (Miller, 1971)*, C. arveloi* and *C. exoptata,* only the proximal receptacle was observed (S3 Table, S14B Fig). The same pattern was observed within other flabellinid taxa, including *Coryphella,* the *Flabellina* s.l. clade, and across Samlidae (Fig 2, S5, S9, S23 Figs). High variation of this feature within these clades limits its usefulness as a distinctive trait to support the separation of monotypic genera.

10] Length of the vas deferens, granulated prostate, and penial morphology. These three features were also used to support the establishment of separate genera within Coryphellidae, Flabellinidae, and Paracoryphellidae [22]. However, our analysis shows their limited phylogenetic significance. Although a long vas deferens is observed in most Paracoryphellidae (S3 Table), in *Mgueolia* it may be long or moderate in length (Fig 4A–C). The occurrence of short and long vas deferens within a single group is also observed within the genera *Coryphella, Samla, Luisella* and the families Flabellinopsidae and Apataidae (S5, S9, S18, S23, S26 Figs; S3 Table). This suggests that the occurrence of a long vas deferens across distant lineages is likely the result of retention of a plesiomorphic state rather than an apomorphic feature, and thus it cannot be used to support the distinct status of a given high-rank taxonomical group. The presence of well-developed, granulated prostatic region was used to support the distinct status of *Chlamylla* from the closely related *Paracoryphella* and *Polaria* [22]*,* however, in *Luisella* this is just intrageneric variation (*Luisella babai* has a granulated prostate, while in other species the prostatic region is indistinct, S23B–D Fig). Finally, most flabellinid species have a conical penis, but in some groups, it can be very different in shape (most diverse in *Coryphella*) (S3 Table). The armed penis of *Unidentia* represents a reliable synapomorphy of this genus, differentiating it from *Pacifia* and *Piseinotecus soussi* (S28 Fig; S3 Table).

## Systematics

**Order Nudibranchia de Blainville, 1814**

**Suborder Cladobranchia William & Morton, 1984**

**Superfamily Fionoidea J. E. Gray, 1857**

**Family Flabellinidae Bergh, 1889**

= Coryphellidae Bergh, 1889

= Paracoryphellidae Miller, 1971

*Diagnosis.* Head rounded or elongate; anterior foot corners present; notal edge well-developed or reduced, continuous or discontinuous, or completely reduced; cerata attached directly on body, on low elevations, or on compound peduncles, in groups or in continuous rows; rhinophores smooth, rugose, papillate, annulate or perfoliate; anus pleuroproctic; oral glands present or absent; jaws triangular with one to several rows of denticles on masticatory border; triserial radula; triangular rachidian teeth with small depressed or elevated conical cusp and numerous denticles on each side; lateral teeth

wide or narrow triangles with well-developed denticles or reduced denticles on inner side and massive or subulate apical cusp, denticles may be absent; one or two seminal receptacles, simple or bilobed, may be absent in some groups; short to long vas deferens with indistinct or well-developed granulated prostate; penial gland absent; unarmed penis.

1. **Genus *Flabellina* McMurtrie, 1831**

   *Flabellina* McMurtrie, 1831: 343.

   = *Calmella* Eliot, 1910: 134. Type species: *Eolis cavolini* Vérany, 1846, by monotypy.

   = *Carronella* Korshunova et al., 2017: 46–47. Type species: *Eolis pellucida* Alder & Hancock, 1843 by original designation, syn. nov.

   = *Paraflabellina* Korshunova et al., 2017: 52. Type species: *Flabellina ischitana* Hirano & Thompson, 1990 by original designation, syn. nov. (S2–S5 Figs)

*Diagnosis*. Body commonly narrow; head rounded to elongate; notal edge completely reduced or reduced, discontinuous; rhinophores smooth or annulate, with 10–19 rings; cerata arranged in groups, on low elevations or compound peduncles; anus pleuroproctic or acleioproctic; cerata translucent; digestive gland visible through the ceratal epidermis; oral glands present; masticatory border denticulated, ranging from one to several rows of denticles; radula triserial; rachidian teeth usually with depressed and compressed conical cusp, followed by 4–10 denticles on each side; lateral teeth wide triangles with reduced denticles, or wide with broad concave base, narrow blunt cusp; denticles present or absent; proximal seminal receptaculum present or absent, simple or bilobed; distal seminal receptacle simple or bilobed, present or absent; short to moderately long vas deferens; penis unarmed.

*Type species: Doris affinis* (Gmelin, 1791), by monotypy.

*Species included: Flabellina affinis* (Gmelin, 1791)*, Flabellina cavolini* (Vérany, 1846)*, Flabellina funeka* Gosliner, Griffiths, 1981*, Flabellina gabinierei* (Vicente, 1975)*, Flabellina gaditana* (Cervera et al., 1987)*, Flabellina enne* (Korshunova et al., 2017)*, Flabellina pellucida* (Alder & Hancock, 1843)*, Flabellina ischitana* Hirano & Thompson, 1990.

The following species are provisionally placed in *Flabellina* based on morphological data, but molecular data is needed for confirmation: *Flabellina alternata* Ortea & Espinosa, 1998*, Flabellina bandeli* (Marcus, 1976)*, Flabellina evelinae* Edmunds, 1989*, Flabellina ilidioi* Calado, Ortea & Caballer, 2005*, Flabellina rubromaxilla* Edmunds, 2015.

*Remarks.* Until recently [28], the validity of the monotypic genus *Calmella* had not been questioned. The reduced and nearly indistinguishable lateral teeth from *C. cavolini* (S3E Fig) rendered this species (and genus) as monoseriate [9,52,53]. However, the molecular and morphological results of [28] led to the synonymizing of the genus *Calmella* as *Flabellina*. Right after and despite the limited geographic and taxonomic sampling, [22] re-established *Calmella* as a valid genus due to "its much less ramified and raised ceratal stalks and presence of smooth rhinophores".

Additionally, *Carronella* was established by [22] for the species *C. pellucida* and *C. enne*, as the absence of ceratal elevations (S2B Fig) and denticles in lateral teeth (S3C, D Fig) were considered a synapomorphy of this genus. Along with *Carronella*, the authors also established the genus *Paraflabellina* to maintain *Flabellina* s. str. and *Calmella* as monophyletic, without mentioning any synapomorphy for this new genus. Although our results recover the same tree topology for this clade (Fig 1 vs. Fig 1, page 77 of [22]), our re-evaluation of morphological characters supports the synonymization of *Calmella*, *Carronella*, and *Paraflabellina* under the name *Flabellina*. True *Flabellina* species are characterized by the combination of lateral teeth morphology (S3 Fig), rhinophores smooth (*F. bandeli, F. cavolini, F. enne, F. gaditana*, and *F. pellucida*) (S2B, D–F Fig) or annulate (*F. affinis, F. alternata, F. evelinae, F. funeka, F. gabinieri, F. illidoi, F. rubromaxilla* and *F. ischitana*) (S2A, C Fig) and cerata grouping on low elevations (in *F. enne* and *F. pellucida*) (S2B Fig) or on compound peduncles (present in the remaining species of the genus) (S2A, C–F Fig) (S3 Table). It should be mentioned that the particular case of *F. enne* and *F. pellucida*, with smooth rhinophores, non-pedunculate cerata may be a retention of

plesiomorphic states, while the reduction of denticles on lateral teeth in their case could be an apomorphic character. Despite the significant variability, the above-mentioned shared features are unique to the genus and support the redefinition of *Flabellina* as a non-monotypic genus, avoiding taxa based on single autopomorphies.

While molecular phylogenies have clarified the monophyly of an expanded *Flabellina* incorporating *Calmella*, *Caronella*, and *Paraflabellina*, distinguishing it from other closely related genera within Flabellinidae s.l. hinges on a combination of specific morphological traits. Furthermore, *Flabellina* is typified by rhinophores that are either smooth or annulate (S2 Fig), a feature that clearly differentiates it from *Coryphellina* (papillate), *Mgueolia* (rugose), and *Launsina* gen. nov. (perfoliate) (Figs 2, 3A–C, 5A, B, 6A–C, 7, S12 Fig). Additionally, the cerata position on compound penducles serves as a critical diagnostic character since, excluding *F. pellucida* and *F. enne*, the species of this genus are the only ones of Flabellinidae that present this character. Although other features, such as radular morphology and certain reproductive system characteristics, former morphological features define *Flabellina* and robustly differentiate it from other genera recognized in this study.

## 2. Genus *Coryphella* J. E. Gray, 1850

*Coryphella* J. E. Gray, 1850: 428.

= *Corrupta* Korshunova et al., 2025: 75. Type species: *Corrupta alexanderi* (Ekimova, 2022) by original designation, syn. nov.

= *Portorchardia* Korshunova et al., 2025: 80–81. Type species: *Portorchardia candela* Korshunova, Fletcher & Martynov, 2025 by original designation, syn. nov. (S6–S9 Figs)

*Diagnosis*. Body narrow to wide; head rounded; notal edge continuous or discontinuous, well-developed or reduced; unstalked cerata attached directly on body, rarely on low elevations, in groups or in continuous rows; rhinophores smooth, rugose, tuberculated, annulated or weakly perfoliated, rarely with bulbous bases; anus pleuroproctic; oral glands present or absent; jaws with single to several rows of denticles; triserial radula; rachidian teeth with well-defined protruding non-compressed non-depressed short or long cusp, cusp slender or broad, rarely cusp slightly compressed by adjacent denticles; lateral teeth wide or narrow triangles with well-developed denticles on inner side, massive apical cusp; distal and proximal seminal receptacles, proximal simple or bilobed, distal unstalked, rarely absent; short to long vas deferens with indistinct prostate; penial gland absent; unarmed penis.

*Type species*: *Eolis rufibranchialis* Johnson, 1832 [=*Coryphella verrucosa* (Sars, 1829)], by subsequent designation by Alder & Hancock (1855).

*Species included*: *Coryphella alexanderi* Ekimova, 2022, *C. amabilis* (Hirano et Kuzirian, 1991), *C. athadona* Bergh, 1875, *C. borealis* Odhner, 1922, *C. browni* Picton, 1980, *C. candela* (Korshunova et al., 2025) comb. nov., *C. capensis* Thiele, 1925, *C. chriskaugei* (Korshunova et al., 2017), *C. cooperi* Cockerell, 1901, *C. falklandica* Eliot, 1907, *C. fogata* (Millen & Hermosillo, 2007), *C. gracilis* (Alder & Hancock, 1844), *C. lineata* (Korshunova et al., 2017), *C. monicae* (Korshunova et al., 2017), *C. nobilis* Verrill, 1880, *C. orjani* (Korshunova et al., 2017), *C. sanamyanae* (Korshunova et al., 2017), *C. trilineata* O'Donoghue, 1921, *C. trophina* (Bergh, 1890), *C. verrucosa* (Sars, 1829), *C. verta* Marcus, 1970.

The following species are provisionally placed in *Coryphella* based on morphological data, but molecular data are needed for confirmation: *Coryphella abei* Baba, 1987, *C. californica* Bergh, 1904, *C. cerverai* (Fischer et al., 2007) comb. nov., *Coryphella dana* (Millen & Hamann, 2006) comb. nov., *C. insolita* (García-Gómez & Cervera, 1989), *C. pallida* Verrill, 1900.

*Remarks.* The genus *Coryphella* is a traditionally established genus but was restricted to a single species, *C. verrucosa* by [22]. Further studies, however, suggested that this genus should include the entire diversity of the Coryphellidae sensu [22], see [34,41]. Our results are in line with the latter works (Fig 1). We also consider recently established monotypic genera *Corrupta* and *Portorchardia,* see [35], as junior subjective synonyms of *Coryphella*.

Our reconsideration of morphological characters suggests that more species currently accepted as *Flabellina* or *Coryphellina* may be affiliated with *Coryphella.* For example, the rare species *Flabellina dana* from the Caribbean region shares several external and internal features with *Coryphella*: i.e., weakly annulate rhinophores (similar to *C. fogata*) and broad triangular lateral teeth with strong denticles and strongly curved cusps [54,55], which are never observed in *Edmundsella* nor *Flabellina* (Fig 2; S3 Table)*.* At the same time, *F. dana* has an elongated head, cerata on low elevations and slightly depressed central cusp of the rachidian teeth, not typical of *Coryphella,* but present in the newly established Pacific genus *Launsina* gen. nov. (see below). Another problematic species is *Flabellina cerverai,* which was recently transferred to the genus *Coryphellina* due to the bilobed proximal seminal receptacle [22]. However, our study suggests that the morphology of the proximal seminal receptacle could be variable within a single genus (e.g., in *Edmundsella* it could be either bilobed or simple, see below), and the presence of papillate rhinophores is the most important diagnostic character of *Coryphellina* (Fig 2). *Flabellina cerverai* has a rounded head, rugose rhinophores, large triangular plates of the lateral teeth with strong and slightly curved cusps, and elevated, slender, non-depressed central cusps in the rachidian teeth [56] – all these features fit the diagnosis of *Coryphella* (Fig 2, S3 Table). Considering the similarities in rhinophoral and lateral teeth morphology, we provisionally place *Flabellina dana* and *F. cerverai* in the genus *Coryphella,* however this decision is only tentative, as several differences are also observed, and molecular data are needed for their correct generic placement.

3. **Genus *Chlamylla* Bergh, 1886**

*Chlamylla* Bergh, 1886: 9–10.

= *Paracoryphella* Miller, 1971: 315. Type species: *Paracoryphella islandica* (Odhner, 1937) by original designation.

= *Polaria* Korshunova et al., 2017: 18–19. Type species: *Coryphella polaris* Volodchenko, 1946 by original designation, syn nov. (S10, S11 Figs)

*Diagnosis.* Body wide; head rounded; notal edge present, well-developed, continuous; rhinophores smooth to rugose; cerata in rows, attached directly to notum, unstalked; anus pleuroproctic at middle of body; oral glands absent; jaws with rows of denticles on masticatory border; triserial radula; rachidian teeth with narrow to broad elongate non-compressed conical elevated cusp with small to large denticles on each side; lateral teeth narrow to wide triangular, smooth to highly denticulate; one distal seminal receptacle; vas deferens long; prostate indistinct or granulated; penial gland absent; unarmed penis, sometimes penial collar present; sometimes penis external, non-retractable.

*Type species: Chlamylla borealis* Bergh, 1886, by monotypy.

*Species included: Chlamylla borealis* Bergh, 1886*, Chlamylla intermedia* (Bergh, 1899), *Chlamylla islandica* (Odhner, 1937) comb. nov., *Chlamylla polaris* (Volodchenko, 1946) comb. nov.

The following species are provisionally placed in *Chlamylla* based on morphological data, but molecular data is needed for confirmation: *Chlamylla ignicrystalla* (Korshunova et al., 2017) comb. nov., *Chlamylla parva* (Hadfield, 1963) comb. nov.

*Remarks.* The genera *Chlamylla, Paracoryphella* and *Polaria* sensu [22] display relatively low molecular divergence (9%–14% between different genera, see [22] and S4 Table) compared to other flabellinid taxa, especially *Coryphellina, Samla* and *Luisella* (Fig 1). In the closely related genus *Ziminella* the values of interspecific p-distances are higher than p-distances between the genera *Chlamylla, Paracoryphella*, and *Polaria* [22,57]. All three genera are characterized by low species numbers (*Polaria* is monotypic), but share specific common traits: wide body, well-developed continuous notal edge, rachidian teeth with large non-compressed elevated cusp, and the presence of a single seminal receptacle (S10, S11 Figs). The differences between genera (granulated prostate of *Chlamylla* (S11A Fig)*,* wide lateral teeth of *Polaria* (S10F Fig), and external non-retractable penis of *Paracoryphella* (S11B Fig)) appear to be autapomorphic features, similarly to that in *Luisella* (granulated prostate in *L. babai,* S23C Fig), *Coryphella* (wide and denticulate lateral teeth in *C. trophina* readily different from

the typical coryphellid pattern, S7E Fig), and *Notaeolidia* (external penis referred as exterior genital lobe in *N. schmeckelae*, see [58]). Based on this viewpoint and considering the low intergeneric p-distances (S4 Table), we suggest the synonymization of the genera *Polaria* and *Paracoryphella* with *Chlamylla*. *Chlamylla* differs from *Mgueolia* in radular morphology (small and depressed central cusp on rachidian teeth in *Mgueolia* and plate-like lateral teeth, Figs 3C, D, G, 5D, E, H, I, 6C, D, G, H) and from *Ziminella* by the presence of a single receptaculum (absent in *Ziminella*, S11D Fig).

4. **Genus *Coryphellina* O'Donoghue, 1929**

*Coryphellina* O'Donoghue, 1929: 797–798

= *Nossis* Bergh, 1902: 52 [non *Nossis* Kinberg, 1865 (Polychaeta)]. Type species: *Nossis indica* Bergh, 1902 by monotypy. (S12–S14A, B Figs)

*Diagnosis*: Body narrow; head elongated; notal edge reduced, discontinuous; cerata in groups on low elevations, posterior cerata sometimes attached directly to body; rhinophores papillate; anus pleuroproctic; oral glands present; jaws with several rows of denticles; triserial radula; rachidian teeth with compressed and depressed cusp and strong denticles on each side; lateral teeth triangular with or without long attenuated distal process at base, with strong or subulate cusp, denticles on inner edge present, sometimes small dentitions on outer edge present; proximal receptacle bilobed; distal receptacle present or absent; short vas deferens with indistinct prostate; penial gland absent; unarmed penis.

*Type species*: *Coryphellina rubrolineata* O'Donoghue, 1929, by monotypy.

*Species included*: *Coryphellina arveloi* (Ortea & Espinosa, 1998), *Coryphellina aurora* Ekimova et al., 2022, *Coryphellina delicata* (Gosliner & Willan, 1991), *Coryphellina exoptata* (Gosliner & Willan, 1991), *Coryphellina flamma* Ekimova et al., 2022, *Coryphellina lotos* Korshunova et al., 2017, *Coryphellina pannae* Ekimova et al., 2022, *Coryphellina pseudolotos* Ekimova et al., 2022, *Coryphellina rubrolineata* O'Donoghue, 1929.

The following species are provisionally placed in *Coryphellina* based on morphological data, but molecular data are needed for confirmation: *Coryphellina albomarginata* (M. C. Miller, 1971), *Coryphellina hamanni* (Gosliner, 1994), *Coryphellina indica* (Bergh, 1902), *Coryphellina marcusorum* (Gosliner & Kuzirian, 1990), *Coryphellina poenicia* (Burn, 1957), *Coryphellina westralis* (Burn, 1964).

*Remarks*. The distinct status of *Coryphellina* was re-established following the revision by [22]. Papillate rhinophores and the bilobed proximal seminal receptacle were considered as the most important diagnostic features of this genus. Our results fully confirm this view, however, the rugose rhinophores of the species *F. cerverai* (considered as a member of *Coryphellina* in [22] and other morphological features do not fit the generic diagnosis, thus this species is transferred to *Coryphella* herein (see above)). Also, it should be noted that the bilobed seminal receptacle could be found in other genera, e.g., *Edmundsella* (S14C Fig), therefore the presence of papillate rhinophores is the most important diagnostic feature of *Coryphellina*. Within the genus *Coryphellina* we identified at least two diverse species complexes: the *Coryphellina rubrolineata* species complex is represented by at least nine species, five of which have been described in previous studies [22,59,60] and the *Coryphellina exoptata* species complex, represented by two species (Fig 1).

5. **Genus *Edmundsella* Korshunova et al., 2017**

*Edmundsella* Korshunova et al., 2017: 51. (S14C, D, S15 Figs)

*Diagnosis*. Body narrow; head rounded to elongate; notal edge reduced discontinuous; cerata in groups on low elevations or rarely attached directly to body; rhinophores smooth; anus pleuroproctic; oral glands present; jaws with single to several rows of denticles; triserial radula; rachidian teeth with compressed and depressed cusp and strong denticles on each side; lateral teeth triangular with concave base and long, slender tip, denticles on inner edge present, large; proximal seminal receptaculum simple or bilobed; distal seminal receptacle simple or absent; short vas deferens with indistinct prostate; penial gland absent; unarmed penis.

*Type species: Edmundsella pedata* (Montagu, 1815), by original designation.

*Species included: Edmundsella albomaculata* (Pola et al., 2015)*, Edmundsella bertschi* (Gosliner & Kuzirian, 1990) comb. nov.*, Edmundsella dushia* (Ev. Marcus & Er. Marcus, 1963) comb. nov., *Edmundsella pedata* (Montagu, 1815), *Edmundsella vansyoci* (Gosliner, 1994).

*Remarks.* The genus *Edmundsella* was described in the revision by [22] for two Atlantic species *Flabellina pedata* and *F. albomaculata* and the Pacific species *Edmundsella vansyoci.* In a subsequent work, *Flabellina bertschi* was identified as a member of the genus *Edmundsella,* but without any additional explanation [61]. Our molecular phylogenetic analysis confirms this view (Fig 1). Also, COI sequences of *Flabellina dushia* are available in BOLD, and our additional analysis indicates this species is also a member of the genus *Edmundsella* (S30 Fig). The primary differences between *Edmundsella* and other flabellinid genera established by [22] were: 1) smooth rhinophores, 2) non-stalked cerata on low elevations, 3) relatively short vas deferens. However, these features also appear across various species of *Coryphella* and *Flabellina* s.l. (see above). Our morphological analysis indicates that all members of *Edmundsella* possess a common rachidian and lateral teeth morphology: the rachidian teeth possess a depressed and small conical cusp, always compressed by adjacent denticles, and the lateral teeth have a concave base and a long, slender tip; denticles on the inner side of the lateral cusp are present and well-developed (S15E–G Fig). These features may be observed in the genus *Launsina* gen. nov. (Fig 7E, F)*,* but are not typical for *Coryphella* (S7 Fig)*.* As mentioned above, *Flabellina* may have a similar configuration of lateral teeth (e.g., *Flabellina ischitana,* S3H Fig), but its species differ from *Edmundsella* by cerata organized in compound peduncles (S2C Fig). It should also be noted that different *Edmundsella* species have a variable number and diverse morphology of seminal receptacles: *E. bertschi, E. pedata* and *E. vansyoci* have a single proximal receptacle, which is bilobed (S14C Fig), whereas in *E. albomaculata* and *E. dushia* both proximal and distal receptacles are observed, but they represent simple muscular sacs (S14D Fig). These data suggest that the morphology of seminal reservoirs may have a significant intrageneric variation, and this feature has a less diagnostic value at the genus-level than it was previously expected [22].

## 6. Genus *Ziminella* Korshunova et al., 2017

*Ziminella* Korshunova et al., 2017: 19–20. (S11D, S16 Figs)

*Diagnosis*. Body wide; head rounded; notal edge present, well-developed, continuous; rhinophores smooth to rugose; cerata in rows, attached directly to notum, unstalked; anus pleuroproctic in second part of body; oral glands absent; jaws with blunt denticles on masticatory border; triserial radula; rachidian teeth with strong elevated cusp, large denticles on each side; lateral teeth narrow, triangular, with minute denticles on inner edge or denticles absent; seminal receptacles absent; vas deferens long; penial gland absent; unarmed penis.

*Type species: Eolis salmonacea* Couthouy, 1838, by original designation.

*Species included: Ziminella abyssa* Korshunova et al., 2017*, Ziminella japonica* (Volodchenko, 1941), *Ziminella salmonacea* (Couthouy, 1838), *Ziminella vrijenhoeki* Valdés et al., 2018

*Remarks.* This genus was recently reviewed by [55]. *Ziminella* represents wide-bodied flabellinids inhabiting muddy sand and silty communities in boreal and Arctic regions from the upper subtidal to abyssal depths [22,33,36,57]. It is well differentiated from the closely related *Mgueolia* and *Chlamylla* by the absence of the seminal receptacle and by the radular morphology (see remarks for the genus *Chlamylla*).

## 7. Genus *Mgueolia* Korshunova, Fletcher & Martynov, 2025

*Mgueolia* Korshunova, Fletcher & Martynov, 2025: 27–28. (3, 4A–C, 5, 6 Figs)

*Diagnosis*. Body moderately wide; head rounded; notal edge present, well-developed, continuous; rhinophores rugose; cerata in rows, attached directly to notum, unstalked; anus pleuroproctic; oral glands absent; jaws denticulate; triserial

radula; rachidian teeth with compressed and depressed slightly protruding central cusp and several large denticles on each side; lateral teeth plate-like triangular, with wide base, attenuate tapering large or narrow cusp and with irregular denticles on inner edge or denticles absent; proximal seminal receptacle present or absent; distal seminal receptacle elongate; vas deferens moderate with or without expanded glandular prostatic part; penial gland absent; unarmed penis.

*Type species: Mgueolia almamater* Korshunova, Fletcher & Martynov, 2025, by monotypy.

*Species included: Mgueolia almamater* Korshunova, Fletcher & Martynov, 2025*, Mgueolia anastasiae* sp. nov.*, Mgueolia vladimiri* sp. nov.*, Mgueolia ninae* sp. nov.

*Remarks.* The genus *Mgueolia* was recently described based on material from the Kurile Islands with *Mgueolia almamater* as a type species [49]. Its original description contained only a brief diagnosis of the most important diagnostic features, and no molecular data were provided for this species [49]. Further work by the same authors [35] included its morphological redescription and testing its phylogenetic placement, which showed that this genus is sister to other paracoryphellid groups: *Ziminella, Chlamylla, Paracoryphella* and *Polaria* [35]. As primary differences of *Mgueolia* from other paracoryphellids researchers identified the radular patterns, i.e., narrow and compressed cusp of the rachidian teeth and plate-like and almost smooth lateral teeth resembling those in the family Eubranchidae.

Our results confirm that *Mgueolia* represents a separate, supported clade in the molecular phylogenetic tree (Fig 1) and it is close to the genera *Chlamylla* and *Ziminella*. From the deep waters off the Kurile Islands, we also discover four distinct species, which differed in morphology and also showed a high molecular divergence in all studied markers (Fig 1, S1 Fig, S5 Table; Data S1). Three of them are described herein, but the material of the fourth species was in poor preservation condition, which did not allow us to study its morphology, and therefore it is referred here as *Mgueolia* sp. 1.

Our results also confirm that *Mgueolia* is well differentiated from *Chlamylla* s.l. (including *Polaria* and *Paracoryphella* sensu [22]) and *Ziminella* by a combination of external and internal features. The most important difference is the morphology of the radula, as it was described in [49]. *Mgueolia* has a depressed central cusp of the rachidian teeth that is compressed by adjacent denticles and only protrudes slightly (Figs 3C, D, G, 5D, E, 6C, D, H), while in *Chlamylla* (S10D–F Fig) and *Ziminella* (S16B, E Fig), the central cusp is large and strongly protracted, and the denticles on each side of the cusp are much smaller than in *Mgueolia.* The lateral teeth of *Mgueolia* (Figs 3D, G, 5E, H, I, 6D, G) are also different from those of *Chlamylla* and *Ziminella,* as in the two other groups, they are either hook-shaped or narrow triangles lacking or bearing small denticles (S10D, E, 16B, E Figs), or wide triangular teeth with strong cusps and strong denticles on the inner edge (as in *Ch. polaris* comb. nov., S10F Fig). Also, fusiform short cerata with voluminous digestive gland inside are not typical in *Chlamylla* and *Ziminella* and represent a useful trait for identification of *Mgueolia* in the field, in addition to a narrower body in *Mgueolia* compared to *Chlamylla* and *Ziminella* (S3 Table).

*Mgueolia vladimiri* sp. nov.

urn:lsid:zoobank.org:act:EE2471E7-8638-4532-8EE4-8D76622D7D45 (Figs 3 and 4A)

*Type material:* Holotype MIMB52148, specimen 10 mm in length, R/V "Academic Oparin", st. 2, North-West Pacific, Sea of Okhotsk, Simushir Is., Diana Strait, 47°15.720 N 152°10.028 E, 221 m in depth, 19.08.2024, coll. Darya Grishina. Paratypes: Same locality, date and collector as holotype; MIMB52146, 1 specimen, 12 mm in length, dissected; MIMB52147, 1 specimen, 9 mm in length.

*Type locality:* North-West Pacific, Sea of Okhotsk, Simushir Is., Diana Strait, 47°15.720 N 152°10.028 E, 221 m in depth.

*Etymology:* This species is named in honor of Vladimir Mordukhovich, PhD, leading scientist at the National Centre of Marine Biology, Far East Branch of Russian Academy of Science for his invaluable contribution in the study of the North-West Pacific nudibranchs. Vladimir Mordukhovich is an organizer and scientific leader of numerous expeditions in the North-West Pacific.

*Description. External morphology* (Fig 3A and B): Body length up to 11 mm. Body moderately wide. Foot moderately wide, anterior corners short, rounded. Oral tentacles wide, short, conical. Rhinophores slightly rugose, nearly smooth, twice as long as oral tentacles. Cerata in distinct rows, attached directly to well-defined notal edge, fusiform to cylindrical, rather thick, lateral cerata much smaller than dorsal ones. Digestive gland diverticula fill almost entire ceratal volume. Anal opening on right side below notal edge, almost on middle part of body. Reproductive opening on right side, on anterior part of body.

*Coloration:* Background body color milky-white to pale peachy. Dorsal side of body, head, rhinophores, oral tentacles densely covered by orange or brownish short stripes or sparse patches. Irregular stripes on rhinophores, oral tentacles, dorsal stripes longitudinal. Cerata covered by same orange sparse pigment at base with distinct subapical orange to brown band on top, with white cnidosac area.

*Internal morphology* (Figs 3C–G, 4A): Jaws massive, oval-triangular with well-developed masticatory border bearing up to 4 rows of narrow sharp denticles. Radular formula: 21 x 1.1.1. Rachidian teeth triangular. Central cusp compressed by adjacent denticles, depressed and slightly protruding. From 5 to 7 large denticles on each side of cusp, denticles of about same size. Lateral teeth plate-like, triangular, with wide almost quadrangular base, attenuate tapering cusp. Denticles absent.

*Reproductive system* (Fig 4A): Ampulla narrow, slightly bent. Vas deferens moderate in length, convoluted, with expanded glandular distal part identifying prostatic region. Penis bulbous. Single distal seminal receptacle, elongated, narrow with distinct narrow duct, entering vagina near its distal end. Vagina moderate in length, slightly bent.

*Distribution.* This species is known only from the type locality.

*Remarks. Mgueolia vladimiri* sp. nov. generally resembles *M. almamater*, the type species of *Mgueolia,* however it has a striped color pattern on the dorsum, which was not described for *M. almamater.* Also, *M. vladimiri* sp. nov. has fewer radular rows (31 in *M. almamater*), and the two innermost lateral denticles on the rachidian teeth are closely placed above the central cusp in *M. almamater*, while in *M. vladimiri* their bases are always separated. Finally, according to the original description, *M. almamater* has both proximal and distal seminal receptacles. Nevertheless, in all our three species, only an elongate distal receptacle was detected (Fig 4A–C). *Mgueolia vladimiri* sp. nov. differs from *M. anastasiae* sp. nov. and *M. ninae* sp. nov. in coloration (it has a distinct striped pattern on the dorsum) and in radular morphology, as its lateral teeth are smooth along their entire length, while the other three species possess minute and reduced denticles on the inner side of the lateral teeth (see below). The minimal uncorrected p-distance between *M. vladimiri* sp. nov. and *M. anastasiae* sp. nov. is 15.7%, between *M. vladimiri* sp. nov. and *A. ninae* sp. nov – 12.8% (S5 Table).

*Mgueolia anastasiae* sp. nov.

urn:lsid:zoobank.org:act:5CE234C4-C707-4634-A2C6-DD558061B939 (Figs 4B and 5)

*Type material:* Holotype MIMB52144, specimen 14 mm in length, dissected, R/V "Academic Oparin", st. 2, North-West Pacific, Sea of Okhotsk, Simushir Is., Diana Strait, 47°15.720 N 152°10.028 E, 221 m in depth, 19.08.2024, coll. Darya Grishina. Paratype: MIMB52145, 1 specimen, 4 mm in length, dissected, same locality, date and collector as holotype.

*Type locality:* North-West Pacific, Sea of Okhotsk, Simushir Is., Diana Strait, 47°15.720 N 152°10.028 E, 221 m in depth.

*Etymology:* This species is named in honor of Anastasia (also transliterated as Anastassya) Maiorova, PhD, leading scientist in National Centre of Marine Biology, Far East Branch of Russian Academy of Science, for her invaluable contribution to the study of the North-West Pacific nudibranchs. Dr. Maiorova was the first person to collect this genus during an expedition to the Kuril Islands in 2019.

*Description. External morphology* (Fig 5A and B): Body length up to 14 mm. Body moderately wide. Foot moderately wide, anterior corners short, rounded. Oral tentacles wide, short, conical. Rhinophores rugose, conical, twice as long as oral tentacles. Cerata in distinct rows, attached directly to well-defined notal edge on dorsal side, fusiform, rather thick,

relatively short; lateral cerata much smaller than dorsal. In juvenile specimen cerata elongate, finger-shaped. Digestive gland diverticula fill almost entire ceratal volume. Anal opening on right side below notal edge, almost on middle part of body. Reproductive opening on right side, on anterior part of body.

*Coloration:* Background body color translucent to milky-white, oral tentacles, rhinophores and cerata of same color, without additional pigmentation. Digestive gland diverticula well-visible through transparent body wall, their color changing from orange at bases of cerata to pale red on apically in larger cerata, diverticula orange in smaller cerata. Cnidosacs white, very small.

*Internal morphology* (Figs 4B, 5C–I)*:* Jaws large, oval-triangular with well-developed masticatory border bearing up to 10 irregular rows of conical blunt denticles. Radular formula: 26 x 1.1.1. Rachidian teeth triangular. Central cusp compressed by adjacent denticles, depressed, slightly protruding. From 4 to 7 large denticles on each side of cusp, denticles of about same size. Lateral teeth plate-like, triangle, with wide and slightly concave base and attenuate tapering triangle cusp. Most laterals bearing irregular elevations or denticles at base of their inner edge, commonly 2–4.

*Reproductive system* (Fig 4B): Ampulla convoluted, with expanded proximal portion, narrowing distally. Vas deferens moderate in length, with expanded proximal portion possessing distinct glandular region likely representing prostate; distal portion of vas deferens narrow. Penis elongated, conical. Single distal seminal receptacle, elongated, narrow with distinct narrow duct, entering vagina near its distal end. Vagina moderate in length, slightly bent.

*Distribution.* This species is known only from the type locality.

*Remarks. Mgueolia anastasiae* sp. nov. resembles *Coryphella nobilis* in external appearance due to its translucent body color, orange-red digestive gland in fusiform cerata and white rugose rhinophores, but clearly differs from the latter by having short conical oral tentacles and a well-developed notal edge. *Mgueolia anastasiae* sp. nov. differs from *M. almamater* and *M. ninae* sp. nov. by its uniform cerata coloration and the reduced denticles on the lateral teeth. Differences from *M. vladimiri* sp. nov. were described above. The minimal uncorrected p-distance between *M. anastasiae* sp. nov. and *M. ninae* sp. nov is 17.6% (S5 Table).

*Mgueolia ninae* sp. nov.

urn:lsid:zoobank.org:act:E7BA4F1F-7657-44E2-A466-DE4ED82DDCFF (Figs 4C and 6)

*Type material:* Holotype MIMB52149, specimen 10 mm in length, dissected, R/V "Academic Oparin", st. 2, North-West Pacific, Sea of Okhotsk, Simushir Is., Diana Strait, 47°15.720 N 152°10.028 E, 221 m in depth, 19.08.2024, coll. Darya Grishina.

*Other material studied*: MIMB52209, 1 specimen, 6 mm in length, only peace of tissue available, R/V "Academic Oparin", st. 3, North-West Pacific, Urup Is., 45°35.5 N 149°47.7 E, 142–145 m in depth, 27.06.2019, coll. Anastassya Maiorova.

*Type locality:* North-West Pacific, Sea of Okhotsk, Simushir Is., Diana Strait, 47°15.720 N 152°10.028 E, 221 m in depth

*Etymology*: This species is named in honor of Nina Izotovna Volodchenko, USSR zoologist, the first specialist in systematics and biodiversity of boreal and Arctic USSR nudibranch molluscs, author of several species descriptions, including *Ziminella japonica* (originally described as *Coryphella japonica*) and *Chlamylla polaris* (originally described as *Coryphella polaris*).

*Description. External morphology* (Fig 6A and B)*:* Body length up to 10 mm. Body moderately wide. Foot moderately wide, anterior corners short, rounded. Oral tentacles elongate, conical. Rhinophores rugose, conical, same size as oral tentacles. Cerata in distinct rows, attached directly to well-defined notal edge, dorsal side, elongate, cylindrical, lateral cerata much smaller than dorsal. Digestive gland diverticula fill almost entire ceratal volume. Anal opening on right side below notal edge, almost on middle part of body. Reproductive opening on right side, on anterior part of body.

*Coloration:* Background body color translucent to milky-white, oral tentacles of same color. Head, rhinophores with slight greenish or pale orange-brown pigmentation. Digestive gland diverticula well visible through transparent body wall, beige to light brown or orange in dorsal cerata, pale beige in lateral small cerata. Cnidosacs white.

*Internal morphology* ([Figs 4C](), [6C–H]()): Jaws massive, oval-triangular with well-developed masticatory border bearing up to 5 rows of conical denticles, outer denticles elongate, curved. Radular formula: 22 x 1.1.1. Rachidian teeth triangular. Central cusp compressed by adjacent denticles, depressed, slightly protruding. From 5 to 7 large denticles on each side of cusp, denticles similar in size. Lateral teeth plate-like, triangular, with wide almost quadrangular base, attenuate tapering cusp. Laterals always bearing from 2 to 8 small irregular denticles at base of inner edge.

*Reproductive system* ([Fig 4C]()): Ampulla convoluted, narrow. Vas deferens moderate in length, with expanded middle portion with distinct glandular region likely representing prostate; proximal, distal portions of vas deferens narrow. Penis conical. Single distal seminal receptacle, elongated, narrow with distinct narrow duct, entering vagina near distal end. Vagina hidden in female gland mass, morphology not observed.

*Distribution.* This species is known only from the type locality.

*Remarks.* Differences of *M. ninae* sp. nov. from two new *Mgueolia* species were described above; from *M. almamater* it differs by lateral teeth, which are always denticulated in *M. ninae* sp. nov. ([Fig 6D]() and [G]()), but commonly smooth in *M. almamater* [49].

## 8. Genus *Launsina* gen. nov.

urn:lsid:zoobank.org:act:2EB50B66-FDBA-4608-9056-19646918ED81 ([Figs 4D](), [7]())

*Diagnosis*: Body narrow; head elongate; notal edge completely reduced; unstalked cerata in groups on low elevations; rhinophores elongate, conical, densely perfoliated; anus pleuroproctic; oral glands present; jaws with up to 3 rows of denticles on masticatory border; triserial radula; rachidian teeth with well-defined very finely depressed cusp, sometimes slightly compressed by adjacent denticles, about same size as denticles; lateral teeth with wide base, narrow subulate cusp, bearing denticles on inner edge; distal, proximal seminal receptacles; short vas deferens with indistinct prostate; penial gland absent; unarmed penis.

*Etymology:* After Launsina (Laun-Sina), the Philippine goddess of the eastern skies, stars and seas who brings in the light of the sun and cool winds during hot days of the dry season, and is also a guardian against strong typhoons.

*Type species: Launsina tanyae* sp. nov., here designated.

*Species included*: *Launsina tanyae* sp. nov., *Launsina rupropurpurata* (Gosliner & Willan, 1991) comb. nov.

*Remarks*: The species *Launsina rubropurpurata* comb. nov. (originally described as *Flabellina rubropurpurata*) was considered to be morphologically and phylogenetically close to species currently belonging to the genus *Samla* (family Samlidae) [19,22]. However, *Launsina rubropurpurata* comb. nov. as well as *Launsina tanyae* sp. nov. differ from *Samla* species by important morphological characters. First of all, all members of the family Samlidae have cerata attached to the body on serial common peduncles ([S22A–E](), [S24A, F Figs]()), while in the genus *Launsina* gen. nov. cerata are grouped on low elevations ([Fig 7A](), [B]()). Also, in members of the genus *Samla* the proximal seminal receptacle is reduced into an expanded region of the oviduct ([S23A Fig]()), and in most representatives of genus *Luisella* the distal receptacle is reduced ([S23B, C Fig]()), while in *Launsina* gen. nov. both the proximal and distal seminal receptacles are present and well-developed ([Fig 4D]()). In *Launsina* gen. nov. the notal edge is completely reduced, in contrast to *Samla* ([S3 Table]()). Among Flabellinidae the genus *Launsina* gen. nov. displays a closer relationship with *Coryphella,* but the latter differs from *Launsina* gen. nov. by the rounded head ([S6 Fig]()) (elongate in *Launsina* gen. nov., [Fig 7A](), [B]()), the morphology of the rhinophores (only in *C. trophina* the rhinophores are very weakly perfoliated), the notal edge morphology (in *Coryphella* it readily presents, but reduced in most species), the ceratal arrangement (in most *Coryphella* cerata are attached to the body), and in the radular morphology ([Figs 2](), [7](), [S6](), [S7 Figs]()). In the majority of *Coryphella* the central cusp is protruding and elevated in contrast to *Launsina* gen. nov.; in all *Coryphella* the lateral teeth form wide or narrow triangles with well-developed cusps

(S7 Fig), while in *Launsina* gen. nov. they have wide bases and slender tips (Fig 7E, F), which is more characteristic of the genus *Edmundsella* (S15E–G Fig). We considered a possible inclusion of members of the genus *Launsina* gen. nov. in *Coryphella* to be an impractical solution, as this renders the diagnosis of *Coryphella* identical to that of *Edmundsella.* Since the members of the genus *Launsina* gen. nov. represent a derived clade in the phylogenetic tree, sister to the genus *Coryphella,* the distinct status of this genus is confirmed by molecular data as well.

The original description of *L. rubropurpurata* comb. nov. was based on material from very distant localities, i.e., Papua New Guinea and South Africa, and thus this species may represent a species complex. Thus, we suggest that an integrative revision should be conducted to confirm or reject the identity of specimens from distant localities. At this point we prefer to avoid designating this species as the type species of the newly described genus. Our specimens from New Caledonia (Fig 7A) perfectly match the external morphology of the original description of *L. rubropurpurata* comb. nov. [19], while the specimen collected from Vietnam differs from in coloration (Fig 7B). This specimen also differs from the New Caledonia *L. rubropurpurata* comb. nov. genetically and thus represents a new species, *Launsina tanyae* gen. et sp. nov. described herein.

*Launsina tanyae* sp. nov.

urn:lsid:zoobank.org:act:D9DD8971-A7DA-4F11-A33A-36E1C3FB8BD8 (Figs 4D, 7B–F)

*Type material:* Holotype MIMB52174, specimen 7 mm in length, dissected, South China Sea, Vietnam, Nha Trang, Nok Is., N 12° 11.360' E 109° 20.841', 20 m in depth, 27.07.2023, coll. Tatiana Antokhina.

*Type locality:* South China Sea, Vietnam, Nha Trang, Nok Is., N 12° 11.360' E 109° 20.841', 20 m in depth.

*Etymology.* This species is named in honor of our friend and colleague Tatiana Antokhina (shortened as Tanya), Russian marine biologist, diver, and the collector of this species, for her invaluable contributions in the collection of nudibranchs during different expeditions to tropical and boreal seas, and studies of their ecology and spatial distribution.

*Description. External morphology* (Fig 7B)*:* Body length up to 10 mm. Body elongate, narrow, tail tapering posteriorly. Notal edge completely reduced. Foot narrow, anterior corners elongated, tentacular, curved. Oral tentacles elongate, narrow. Rhinophores densely perfoliate with 13 lamellae, conical, twice as short as oral tentacles. Cerata in distinct groups, placed on low elevations, short, fusiform. First ceratal group with up to 7 cerata, 4 cerata in second group and 2 cerata in other groups. Digestive gland diverticula fill almost entire ceratal volume. Reproductive opening on right side, below first group of cerata. Anus pleuroproctic, located between first and second groups of cerata.

*Coloration* (Fig 7B)*:* Body uniformly pink purple with very sparse white speckles covering notal margin, dorsal side of posterior body part. Oral tentacles proximally magenta, white opaque tips. Rhinophores with white bases, yellow middle, orange tips. Cerata bright orange with red cnidosacs.

*Internal morphology* (Figs 4D, 7C–F)*:* Oral glands present, branched. Jaws delicate, oval-triangular, well-developed masticatory border bearing 2 rows of small denticles. Radular formula: 24 x 1.1.1. Rachidian teeth triangular. Central cusp slightly compressed, depressed, elongated. Seven denticles on each side of central cusp. Lateral teeth triangular with concave bases, long, slender tips, 4–6 denticles on inner edge of teeth.

*Reproductive system* (Fig 4D): Ampulla voluminous proximally, narrowing, twisted distally. Vas deferens short, slightly bent, with distal expanded portion before penial sac. Prostate indistinct. Penis conical. Distal and proximal seminal receptacles, proximal smaller with well-defined duct, distal large muscular sac. Vagina short, narrow.

*Distribution.* This species is known only from the type locality. However, specimens with similar coloration are known from Japan [62].

*Remarks.* This species differs from *L. rubropurpurata* comb. nov. in color pattern (Fig 7A, B). In *L. rubropurpurata* comb. nov. (Fig 7A) the dorsal side of the body is covered with dense white opalescent speckles, while in *L. tanyae* sp. nov. this pigmentation is sparse and covers only the notal margin and the posterior end of the body (Fig 7B). The background color is also remarkably different: deep purple in *L. rubropurpurata* comb. nov.*,* and pink purple in *L. tanyae* sp. nov. Finally, the

rhinophores are white with orange tips in *L. rubropurpurata* comb. nov., while in *L. tanyae* sp. nov. they are white proximally, yellow in the middle, and have orange tips. No differences in internal morphology were detected. The uncorrected p-distance between *L. tanyae* sp. nov. and *L. rubropurpurata* comb. nov. is 12.1% (S5 Table). Differences were also observed in all studied nuclear markers (Data S1).

**Family Flabellinopsidae Korshunova et al., 2017**

*Diagnosis.* Head elongate; body moderately wide; anterior foot corners present; notal edge reduced discontinuous; cerata in groups on wide arc-shaped elevations; rhinophores perfoliate or with ridges, granules; anus pleuroproctic or mixed; oral glands present or absent; jaws oval-triangular to oval-rectangular with several rows of denticles on masticatory border; uniserial to triserial radula; triangular or elongated rachidian teeth with non-compressed or compressed small elevated cusp, small denticles on each side; lateral teeth present or absent; lateral teeth wide triangular or elongated, with strong denticles on inner edge or with reduced denticles, outer denticles always present; proximal seminal receptacle present. one or two; short to moderately long prostatic or non-prostatic vas deferens; penial gland absent; unarmed penis.

1. **Genus *Flabellinopsis* MacFarland, 1966**

   *Flabellinopsis* MacFarland, 1966: 308. (S17, 18A Figs)

*Diagnosis.* Body moderately wide; head rounded; notal edge reduced discontinuous; cerata in groups on arc-shaped elevations; rhinophores perfoliate; anus pleuroproctic; oral glands absent; jaws with several rows of denticles; radula triserial; rachidian teeth triangular with non-compressed small cusp, strong denticles on each side; lateral teeth wide, triangular, with long outer basal process, strong conical cusp, large denticles on inner edge, outer denticles present; proximal seminal receptacle serial; long vas deferens with indistinct prostate; penial gland absent; unarmed penis.
   *Type species: Aeolis iodinea* Cooper, 1863 by original designation.
   *Species included: Flabellinopsis iodinea* (Cooper, 1863)

2. **Genus *Baenopsis* Korshunova et al., 2017**

   *Baenopsis* Korshunova et al., 2017: 24 (S18B, S19 Figs)

*Diagnosis*. Body moderately wide; head rounded; notal edge reduced discontinuous; cerata in groups on broad elevations; rhinophores with numerous ridges, granules; anus pleuroproctic; oral glands absent; jaws with several rows of denticles; triserial radula; rachidian teeth triangular with non-compressed small cusp, strong denticles on each side; lateral teeth wide, triangular, with reduced tiny denticles on inner side, with massive apical cusp; small dentitions on outer edge present; proximal seminal receptacle; long vas deferens with wide prostate; penial gland absent; unarmed penis.
   *Type species: Flabellina baetica* García-Gómez, 1984 by original designation.
   *Species included: Baenopsis baetica* (García-Gómez, 1984)
   *Remarks:* The reexamination of internal morphology of *Baenopsis baetica* revealed the presence of outer denticles on the lateral teeth (S19E, F Fig), which have never been observed before [63,64].

3. **Genus *Kynaria* Korshunova et al., 2017**

   *Kynaria* Korshunova et al., 2017: 66–67. (S18C, S20 Figs)

*Diagnosis.* Body moderately wide; head rounded; notal edge reduced discontinuous; cerata in groups on arc-shaped elevations; rhinophores perfoliate; anus pleuroproctic in higher acleioproctic position; oral glands absent; jaws with several rows of denticles; rachidian teeth elongated arc-shaped slightly compressed small cusp, strong denticles on each side; lateral teeth elongated, narrow, curved, with strong curved cusp, large denticles on inner edge, outer denticles present; two proximal seminal receptacles; long vas deferens with wide prostate; penial gland absent; unarmed penis.

*Type species: Coryphella cynara* Marcus & Marcus, 1967 by original designation.
*Species included: Kynaria cynara* (Marcus & Marcus, 1967)

4. **Genus *Bajaeolis* Gosliner & Behrens, 1986**

*Bajaeolis* Gosliner & Behrens, 1986: 106. (S18D, S21 Figs)

*Diagnosis.* Body moderately wide; head rounded; notal edge reduced discontinuous; cerata in groups on arc-shaped elevations; rhinophores perfoliate; anus pleuroproctic, under second ceratal cluster in higher position; oral glands present; jaws rectangular, with few rows of denticles; radula uniserial; rachidian teeth triangular with non-compressed small cusp, strong denticles on each side; lateral teeth absent; proximal seminal receptacle; short vas deferens with indistinct prostate; penial gland absent; unarmed penis.
*Type species: Bajaeolis bertschi* Gosliner & Behrens, 1986 by original designation
*Species included: Bajaeolis bertschi* Gosliner & Behrens, 1986

**Family Samlidae Korshunova et al., 2017**

*Diagnosis.* Head elongate; body narrow; anterior foot corners present or reduced; notal edge reduced discontinuous or absent; cerata in groups on serial peduncles; rhinophores perfoliate or rarely rugose; anus pleuroproctic; oral glands present; jaws triangular with several rows of denticles on masticatory border; triserial radula; triangular rachidian teeth with elevated or slightly depressed small conical cusp, numerous denticles on each side; lateral teeth wide triangular with long outer basal process, strong denticles on inner edge, or denticles absent; one or two seminal receptacles; short to moderately long prostatic or non-prostatic vas deferens; penial gland absent; unarmed penis.

1. **Genus *Samla* Bergh, 1900**

*Samla* Bergh, 1900: 237. (S22, S23A Figs)

*Diagnosis.* Body narrow; head elongate; notal edge reduced discontinuous; cerata in groups on serial peduncles, 2–3 cerata per group; rhinophores perfoliate; anus pleuroproctic; oral glands present; jaws with several rows of denticles; triserial radula; rachidian teeth triangular, with non-compressed elevated or slightly depressed central cusp, numerous denticles with long furrows on each side; lateral teeth wide triangular with long outer basal process, strong inner denticles; distal seminal receptacle present, proximal seminal receptacle serial, representing expanded portion of oviduct, or absent; short to moderately long vas deferens with indistinct prostate; penial gland absent; unarmed penis.
*Type species: Samla annuligera* Bergh, 1900 by monotypy.
*Species included: Samla bicolor* (Kelaart, 1858), *Samla bilas* (Gosliner & Willan, 1991), *Samla macassarana* (Bergh, 1905), *Samla riwo* (Gosliner & Willan, 1991), *Samla takashigei* Korshunova et al., 2017.
*Remarks:* Korshunova et al. [22] reinstated the genus *Samla* for species phylogenetically close to Bergh's *Samla annuligera* [65] (currently considered a junior synonym of *Samla bicolor*). They identified several differences between *Samla* and other Flabellinidae: 1) absence of compound stalks, often non-compressed cusp of the rachidian teeth, short vas deferens. Korshunova et al. [22] also included *Flabellina telja* (considered here as *Luisella telja*) and *F. rubropurpurata* (designated here as *Launsina rubropurpurata* comb. nov.) in the genus *Samla*. Although our analysis supports the distinct status of *Samla,* it also shows that neither *F. telja,* nor *F. rubropurpurata* are the part of this group (Fig 1). The position of *F. rubropurpurata* has a crucial importance, as it questions the differences between the diagnoses of *Samla* and Flabellinidae suggested by [22]. Our morphological analysis instead shows that the primary difference between the Flabellinidae and Samlidae is the ceratal arrangement: in the latter family their arrangement on the common elevation or peduncle is serial: the size of the cerata gradually increases from lateral to dorsal (Fig 3, S22A–E, S24A, F Figs), this feature is not known for any species of Flabellinidae s.str, including the genus *Launsina* gen. nov. (Fig 7A, B). Also, our analysis shows

that *Samla* and *Luisella* represent two distinct genera, which are well differentiated by the number of cerata on a common peduncle (2–3 in *Samla,* 4–8 in *Luisella,* S22A–E, S24A, F Figs). Finally, the notal edge in *Samla* is still present but reduced and discontinuous, whereas it is completely absent in all *Luisella* species studied herein.

Our analysis also reveals a considerable hidden diversity within *Samla,* as we identified several phylogenetically distinct groups likely representing separate species in the *Samla bicolor* species complex (Fig 1). Some of them were collected from Hawaii (S1 Table), which is the type locality of species *Samla annuligera* and *Samla alisonae* Gosliner, 1980*.* Both species are currently considered junior synonym of *Samla bicolor,* and therefore our results suggests that the distinct status of some of these species should be further reconsidered.

## 2. Genus *Luisella* Korshunova et al., 2017

*Luisella* Korshunova et al., 2017: 61. (S23B–D, S24 Figs)

*Diagnosis.* Body narrow; head elongate; notal edge completely absent; cerata in groups on serial peduncles, 4–8 cerata per group; rhinophores perfoliate or rugose; anus pleuroproctic; oral glands absent; jaws with several rows of denticles; triserial radula; rachidian teeth triangular with small conical compressed non-depressed or slightly depressed cusp, denticles on each side; lateral teeth wide triangular with long outer basal process, denticles strong inner or absent; distal seminal receptacle present or absent, proximal seminal receptacle simple; short to moderately long vas deferens with indistinct or granulated prostate; penial gland absent; unarmed penis.

*Type species: Flabellina babai* Schmekel, 1972, by monotypy.

*Species included: Luisella babai* (Schmekel, 1972), *Luisella bulbosa* (Ortea & Espinosa, 1998) comb. nov., *Luisella engeli* (Marcus & Marcus, 1968), *Luisella telja* (Marcus & Marcus, 1967).

The following species are provisionally placed in *Luisella* based on morphological data, but need the molecular data for confirmation: *Luisella llerae* (Ortea, 1989) comb. nov.

*Remarks:* The reconstructed phylogenetic relationships (Fig 1) show that the species *Flabellina bulbosa, Flabellina engeli*, and *Samla telja* (all generic names sensu [22]) clustered with *Luisella babai.* The latter species was placed in the separate genus *Luisella,* because it has a granulated prostate [22]. None of the remaining closely-related species has this type of prostate, however all of them have a common pattern of the ceratal arrangement: serial peduncles with 4–8 cerata per group, which differentiates all these species from *Samla.* This, along with molecular divergence, support the distinctness of the genus *Luisella.* Most of *Luisella* species also have perfoliated rhinophores, except *L. bulbosa,* in which rhinophores are rugose (S3 Table).

## Family Apataidae Korshunova et al., 2017

*Diagnosis.* Head elongate; anterior foot corners present; notal edge completely reduced; cerata arranged in groups, placed on comb-shaped peduncles; rhinophores annulate or wrinkled; anus pleuroproctic in higher acleioproctic position; oral glands present; jaws triangular with one or two rows of denticles on masticatory border; triserial radula; triangular rachidian teeth with small conical compressed non-depressed cusp, numerous denticles on each side; lateral teeth wide triangular with long outer basal process, smooth; one distal seminal receptacle, stalked; short to long vas deferens with indistinct prostate; penial gland absent; unarmed penis.

## 1. Genus *Apata* Korshunova et al., 2017

*Apata* Korshunova et al., 2017: 64. (S25, S26A Figs)

*Diagnosis.* Body narrow; head elongate; notal edge completely reduced; cerata arranged in groups, placed on comb-shaped peduncles; rhinophores annulate; anus pleuroproctic in higher acleioproctic position; oral glands present; jaws triangular with one row of denticles on masticatory border; triserial radula; triangular rachidian teeth with small conical

compressed non-depressed cusp, numerous denticles on each side; lateral teeth wide triangular with long outer basal process, smooth; distal stalked seminal receptacle; long vas deferens with indistinct prostate; penial gland absent; unarmed penis.

*Type species: Coryphella pricei* MacFarland, 1966 by original designation.

*Species included: Apata pricei* (MacFarland, 1966).

*Remarks*: According to our phylogenetic analysis (Fig 1) and preliminary morphological analysis (S25 Fig), the genus *Apata* likely includes at least three distinct species, however more material from the NE Pacific should be studied to complete a needed comprehensive revision.

2. **Genus *Tularia* Burn, 1966**

*Tularia* Burn, 1966: 26. (S26B Fig)

*Diagnosis.* Body narrow; head elongate; notal edge completely reduced; cerata arranged in groups, placed on comb-shaped peduncles; rhinophores wrinkled; anus pleuroproctic in higher acleioproctic position; oral glands present; jaws triangular with two rows of denticles on masticatory border; triserial radula; triangular rachidian teeth with small conical compressed non-depressed cusp, numerous denticles on each side; lateral teeth wide triangular with long outer basal process, smooth; one stalked seminal receptacle; short vas deferens with indistinct prostate; penial gland absent; unarmed penis.

*Type species: Cuthona bractea* Burn, 1962 by original designation

*Species included: Tularia bractea* (Burn, 1962)

*Remarks:* The limited distribution of this species (New Zealand: [66]) restricts its availability for research. A specimen from the Victoria Museum (F113298) was loaned for molecular analyses, as morphological studies were refused. However, amplification was unsuccessful, so this data could not be included in our phylogenetic reconstruction.

**Family Notaeolidiidae Eliot, 1910**

*Diagnosis.* Head rounded; body wide; anterior foot corners present; notal edge well-developed, continuous; cerata arranged in continuous rows, attached directly to body; rhinophores annulate or with irregular ridges; anus pleuroproctic; oral glands absent; jaws oval-triangular, smooth; polyserial radula; rachidian teeth triangular or elongated, non-compressed or compressed small elevated cusp, small denticles on each side; lateral teeth present, wide triangular or elongated, with strong denticles on inner edge or with reduced denticles, with or without outer denticles; separate seminal receptacle; short to moderately long prostatic or non-prostatic vas deferens; penial gland absent; unarmed penis.

**1. Genus *Notaeolidia* Eliot, 1905**

*Notaeolidia* Eliot, 1905: 520. (S26C, S27 Figs)

*Diagnosis.* Same as for the family.

*Type species: Notaeolidia gigas* Eliot, 1905 by original designation.

*Species included: Notaeolidia depressa* Eliot, 1905, *Notaeolidia gigas* Eliot, 1905, *Notaeolidia schmekelae* Wägele, 1990.

*Remarks:* Two specimens of *Notaeolidia depressa* were included in the phylogenetic analysis (Fig 1), which displayed high levels of genetic divergence. These data suggest the existence of the cryptic diversity within this group.

**Family Piseinotecidae Edmunds, 1970**

= Unidentiidae Millen & Hermosillo, 2012 syn. nov.

*Diagnosis.* Body narrow; head elongate; notal edge completely reduced; cerata arranged in groups, placed on low elevations or simple peduncles; rhinophores smooth; anus pleuroproctic or pleuroproctic in higher position, sometimes acleioproctic; oral glands commonly present; jaws elongate triangular or rectangular, with one to several rows of denticles on masticatory border or smooth; uniserial radula; triangular to almost rectangular rachidian teeth with narrow to broad non-compressed elevated, short conical cusp, numerous denticles on each side; lateral teeth absent; one or two seminal receptacles; short to moderately long vas deferens with or without distinct prostate region; penial gland absent or present; penis armed or unarmed.

*Remarks. Piseinotecus soussi* clustered with the genera *Pacifia* and *Unidentia* representing the sister group to the latter genus (Fig 1). The high levels of divergence between *P. soussi* and both *Unidentia* and *Pacifia* suggests it may belong to a separate genus within the family Unidentiidae, and this placement could indicate that the family Piseinotecidae may represent a senior synonym of the Unidentiidae. Moreover, the diagnosis of the genus *Piseinotecus* sensu [67] —and of the family Piseinotecidae—updated by [68] fully agrees with the diagnosis of the family Unidentiidae by [22] as follows: 1) uniserial short radula, 2) strong central cusp of the rachidian teeth, 3) a single row of denticles on masticatory process, 4) cerata in groups on low elevations, 5) smooth rhinophores, 6) acleioproctic anus position 7) proximal seminal receptacle. However, the type species of the genus *Piseinotecus, P. divae* is a rare species that has not been studied since its original description [67], and therefore, there is no molecular data available. The morphology of *P. divae* was described in detail, and it fully corresponds to the diagnosis of Unidentiidae. While the available evidence supports the synonymy of Unidentiidae and Piseinotecidae, caution is warranted due to the existence of numerous examples of rampant homoplasies and convergence in aeolid systematics, e.g., [69,70]. At the same time, retaining separate family names for Piseinotecidae and Unidentiidae due to the unavailability of molecular data for *P. divae*, seems to be an impractical solution for several reasons. First, it requires the introduction of a new genus for *P. soussi,* whose diagnosis would be identical to that of *Piseinotecus* s.str. Second, the genus or family affiliations of any new taxon with a piseinotecid-like morphology could not be possible without molecular data. This would create substantial confusion and uncertainty in the entire classification scheme. Therefore, we suggest tentatively regarding the family Piseinotecidae as a senior synonym of the family Unidentiidae, with the caveat that unravelling the phylogenetic position of the type species (*P. divae*) is crucial for the confirmation of this decision. Molecular data for the rest *Piseinotecus* species are also needed to confirm the monophyly of this enigmatic genus.

### 1. Genus *Piseinotecus* Marcus, 1955

*Piseinotecus* Marcus, 1955: 176. (S28A–D Fig)

*Diagnosis.* Body narrow; head elongate; notal edge completely reduced; cerata arranged in groups, placed on low elevations or simple peduncles; rhinophores smooth; anus acleioproctic; oral glands not described; jaws elongate triangular with a single row of denticles; uniserial radula; triangular to broad rachidian teeth with narrow to broad non-compressed conical cusp elevated cusp with denticles on each side; lateral teeth absent; proximal seminal receptacle present; distal seminal receptacle present or absent; short vas deferens with wide, distinct prostate; penial gland absent; penis unarmed.

Type species: *Piseinotecus divae* Er. Marcus, 1955 by original designation.

*Species included*: *Piseinotecus divae* Marcus, 1955, *Piseinotecus soussi* Tamsouri et al., 2014.

The following species are provisionally placed in *Piseinotecus* based on morphological data, but need the molecular data for confirmation: *Piseinotecus ernestina* Ortea & Moro, 2020, *Piseinotecus gonja* Edmunds, 1970, *Piseinotecus kima* Edmunds, 1970, *Piseinotecus minipapilla* Edmunds, 2015; *Piseinotecus sphaeriferus* (Schmekel, 1965).

*Remarks:* The genus *Piseinotecus* differ from the genus *Pacifia* by smooth central cusp of the rachidian tooth, and from the genus *Unidentia* – by unarmed penis.

2. **Genus *Pacifia* Korshunova et al., 2017**

*Pacifia* Korshunova et al., 2017: 56 (S28E Fig)

*Diagnosis.* Body narrow; head elongate; notal edge completely reduced; cerata arranged in groups, placed on low elevations; rhinophores smooth; anus pleuroproctic towards acleioproctic position; oral glands present; jaws elongate triangular with a single row of denticles or weak tubercles; uniserial radula; triangular to broad rachidian teeth with narrow to broad non-compressed conical cusp elevated cusp with denticles on each side; cusp of rachidian tooth bearing denticles, or smooth; lateral teeth absent; two seminal receptacles; short vas deferens with wide, distinct prostate; penial gland present; penis unarmed.

*Type species: Pacifia amica* Korshunova et al., 2017 by original designation.

*Species included: Pacifia amica* Korshunova et al., 2017; *Pacifia goddardi* (Gosliner, 2010)

*Remarks:* According to our molecular phylogenetic analysis *P. goddardi* did not form a monophyletic group (Fig 1), so the identity of this species should be further tested with additional molecular and morphological data.

3. **Genus *Unidentia* Millen & Hermosillo, 2012**

*Unidentia* Millen & Hermosillo, 2012: 155. (S28F, S29 Figs)

*Diagnosis.* Body relatively narrow; head elongate; notal edge completely reduced; cerata arranged in groups, placed on low elevations; rhinophores smooth; anus pleuroproctic towards acleioproctic position; oral glands present; jaws elongate triangular with a single row of denticles or weak tubercles; uniserial radula; triangular to broad rachidian teeth with narrow to broad non-compressed conical cusp elevated cusp with denticles on each side; lateral teeth absent; one or two proximal seminal receptacles; short vas deferens with distinct wide prostate; penial gland absent; penis armed.

*Type species: Unidentia angelvaldesi* Millen & Hermosillo, 2012 by original designation.

*Species included: Unidentia angelvaldesi* Millen & Hermosillo, 2012, *Unidentia aliciae* Korshunova et al., 2019, *Unidentia nihonrossija* Korshunova et al., 2017, *Unidentia sandramillenae* Korshunova et al., 2017.

*Remarks:* Our analysis (Fig 1) and the results of [71] suggest that at least two additional undescribed species (from Australia and Vietnam) could be identified within *Unidentia*.

## Discussion

### Supported divergence of Apataidae, Samlidae and reconsideration of Piseinotecidae

Our molecular phylogenetic analysis confirms that the traditional Flabellinidae s.l. is polyphyletic, generally corroborating the revision by [22]. Representatives of the families Apataidae, Samlidae and Piseinotecidae form three highly supported and derived clades, not related to the 'core' group of Flabellinidae nor to each other (Fig 1, [22]). We tentatively suggest synonymizing Unidentiidae and Piseinotecidae, but this should be reconsidered once molecular data for the type species *P. divae* become available.

We have identified several morphological synapomorphies for the families Apataidae, Samlidae and Piseinotecidae (Fig 2). Some of them have been already discussed (comb-shaped peduncles of Apataidae, the uniserial radula in combination with proximal seminal receptacle of Piseinotecidae, see [22]). Other synapomorphies are newly established; these are perfoliate rhinophores and serial peduncles of Samlidae, a completely reduced notal edge in *Luisella* in contrast to *Samla,* and a serial proximal seminal receptacle of *Samla*. We consider that these synapomorphies should facilitate the correct placement of newly described taxa, even if molecular data are not available. At the same time, the species composition of some genera needs further revision as several diverse species complexes have been identified (see above).

Furthermore, contrary to previous studies, the expected sister relationship of Cumanotidae Odhner, 1907 and Pseudovermidae Thiele, 1931 [72] was not recovered. In both analyses, the family Apataidae was sister to Cumanotidae (Fig 1, S1 Fig), but received no statistical support. The type species *Cumanotus beaumonti* (Eliot, 1906) has a triserial radula [72], but its morphology differs significantly from that of the family Apataidae (S25 Fig) and most taxonomic

characters of Cumanotidae (position of the rhinophores, position of the anus, ceratal grouping) do not correspond the diagnostic features of Flabellindiae s.l. and, in particular, Apataidae. This relationship, however, should be considered as tentative, since our analysis lacks many aeolid taxa, including lineages of the large family Facelinidae s.l. Further resolution of the family-level phylogeny of Aeolidoidea would benefit from the use of genomic data rather than a handful of individual markers, as was already shown for cladobranch molluscs [73].

## New taxa of Flabellinopsidae

One of the most surprising results of the present study is the close relationship of the genus *Bajaeolis* and representatives of the family Flabellinopsidae (Fig 2). *Bajaeolis* has a uniserial radula (S21E–F Fig) and was originally placed in the family Facelinidae [51]. In all our phylogenetic analyses, *Bajaeolis* is recovered in a clade including representatives of the family Flabellinopsidae (Fig 1). In addition, the monotypic genus *Kynaria*, established for the species *Flabellina cynara* and provisionally placed in "Flabellinidae *incertae sedis*" [22], is sister to the clade comprising Flabellinopsidae + *Bajaeolis* (Fig 1). The close relationship of these four monotypic genera is quite unexpected, since they differ significantly in radular morphology. In *Flabellinopsis* and *Baenopsis*, the lateral teeth are wide, triangular, and bear denticles along the inner edge (S17G, H, S19E, G Figs), while in *Kynaria* the lateral teeth are narrow and curved (S20G Fig), and in *Bajaeolis* they are reliably absent (S21E–G Fig). The lateral teeth of *Flabellinopsis, Baenopsis* and *Kynaria* bear denticles along the outer edge (S17H, S19F, S20I Figs), which may be an important synapomorphy of the group. The rachidian teeth of all four species differ in shape, and also in the number and arrangement of denticles (S17G, S19D, S20G, S21F, G Figs). At the same time, all four genera are characterized by a similar arrangement of cerata: they are arranged in wide groups, located on small horseshoe-shaped low elevations (S17A–C, S19A, S20C, S21A Figs). In addition, three of the four genera (*Flabellinopsis, Kynaria, Bajaeolis*) are characterized by having perfoliated rhinophores (Fig 2, S17A–C, S20A–D, S21A–C Figs). Although the rhinophores of *Baenopsis* differ significantly as they highly folded and granulated (Fig 2, S19A Fig), it can be hypothesized that this morphology is derived from perfoliated rhinophores, rather than being formed *de novo* from smooth or wrinkled ones. Taking this assumption into account, four synapomorphies can be established for this group: 1) rachidian teeth with a small protruding central cusp, 2) lateral teeth bearing outer denticles, 3) perfoliated rhinophores, 4) cerata are placed on wide arc-shaped elevations. The fundamental differences in the morphology of the radula between *Flabellinopsis, Baenopsis, Kynaria, Bajaeolis*, and the mosaic pattern of other internal and external traits, together with high genetic distances, suggest the need to retain four monotypic genera within the family Flabellinopsidae. More importantly, although the family Flabellinopsidae forms a sister group to Flabellinidae in the BI analysis (Fig 1, S1 Fig), this relationship was not statistically supported. This, on the one hand, supports the separate status of Flabellinopsidae as different from Flabellinidae and indicates the need for further studies using a genomic approach and an increased sample size.

## Newly described taxa

In the case of the 'core' group of Flabellinidae s.l. (i.e., the families Paracoryphellidae, Flabellinidae, Coryphellidae sensu [22]), our analyses show significant similarities with the results obtained by [22]. However, several important discrepancies at high taxonomic levels were also identified. We have shown that the species *Samla rubropurpurata* sensu [22] forms a sister clade to the genus *Coryphella* (Fig 1) and is described herein as *Launsina* gen. nov. Furthermore, *Launsina* gen. nov. includes a species complex of at least two pseudocryptic species. These species differ from *Coryphella* in several features: the cerata arranged in groups on low elevations, rather than being attached directly to the body (Fig 7A, B), as in most *Coryphella*, and the margin of the notum is completely reduced (S6 Fig). The main difference between *Launsina* gen. nov. and *Coryphella* is the morphology of the lateral teeth of the radula (Fig 7D–F, S7 Fig). *Launsina* gen. nov. has a wide base with an elongated and narrow, rather subulate tip, and the inner part of teeth forms an almost right angle (Fig 7E, F), which is very similar to the lateral teeth of *Edmundsella* (S15E–G Fig). At the same time, lateral teeth of *Coryphella*

are always triangular in shape, almost always slightly curved, and the tip is a well-defined strong cusp (S7 Fig). Thus, *Launsina* gen. nov. shows several features that do not correspond to *Coryphella* diagnosis, and inclusion of these two species into the *Coryphella* will further make its diagnosis identical to that of *Edmundsella*. At the same time, *Edmundsella* cannot be united with *Coryphella,* as they do not represent a monophyletic group because of the position of well-established and easily diagnosed genus *Coryphellina* (Fig 1, S1 Fig). As *Launsina* gen. nov. forms a separate clade in the phylogenetic tree (Fig 1, S1 Fig), we propose that it represents a distinct genus of Flabellinidae s.l.

Notably, the initial placement of *Launsina rubropurpurata* comb. nov. in the genus *Samla* [22] was probably due to the significant similarity of these groups in external morphology (narrow body, perforated rhinophores, a small number of cerata arranged in distinct groups). Nevertheless, in *Launsina* gen. nov. the cerata are located on small elevations and are disordered, while in representatives of Samlidae the arrangement of cerata on a common peduncle is serial: the size of cerata gradually increases from lateral to dorsal (S22A–E, S24A, F Figs). In addition, in *Launsina* gen. nov. the reproductive system is generally similar to that *Coryphella* as they have both distal and proximal seminal receptacles with well-defined ducts (Fig 4D, S6 Fig), whereas in *Samla*, the proximal receptacle is 'serial' [19] and represents a slight expansion of the oviduct (S23A Fig).

Based on material collected from the Kuril Islands, we discovered three new species of the recently described genus *Mgueolia*, which now includes four species (Figs 1, 3, 5, 6). This group is characterized by a unique combination of morphological features for Flabellinidae s.l. (S3 Table): *Mgueolia* and closely related *Ziminella* and *Chlamylla*, have a continuous notal margin, and their cerata are arranged in rows. But in the case of *Mgueolia* the notal margin is much less developed (Figs 3A, B, 5A–C, 6A, B) than in *Chlamylla* and *Ziminella* (S10A, C, S16A, D Figs) and more similar to the reduced but continuous margin of *Coryphella nobilis* (S6C Fig). The anus in *Mgueolia* is located on the posterior half of the body, which is similar to *Ziminella* (S16A, D Fig). At the same time, the radular morphology of *Mgueolia* differs significantly from the closely related *Ziminella* and *Chlamylla*: the rachidian teeth have a significantly depressed central cusp (Figs 3D, G, 5E, 6D, H) (similar to *Edmundsella* and *Coryphellina*, S13, S15E–G Figs); the lateral teeth of *Mgueolia* are substantially flattened and lack or have few denticles (Figs 3D, G, 5E, H, I, 6D, G), and resemble the flattened lateral teeth of the genus *Eubranchus* (Fionidae s.l., see [74]) instead of the typical triangular denticulated lateral plates of Flabellinidae s.l. (see S3, S7 Figs). Our phylogenetic reconstruction shows that species of *Mgueolia* form a monophyletic group close to *Chlamylla* and *Ziminella*. However, these relationships were not supported in the ML analysis (S1 Fig).

### Redefinition of Flabellinidae

Our analyses demonstrate that adding new taxa to the phylogenetic dataset required reassessing genus-level diagnostic characters in Coryphellidae and Flabellinidae sensu [22]. This is particularly true for the placement of the genus *Launsina* gen. nov. Since this genus shares several diagnostic features with *Coryphella* and *Edmundsella,* its inclusion in the family Coryphellidae (in accordance with its phylogenetic relationships) renders the diagnosis of Coryphellidae identical to that of Flabellinidae sensu [22]. The same is true for *Mgueolia* – with the expansion of the diagnosis of Paracoryphellidae to include the diagnostic features of *Mgueolia*, it largely overlaps with that of Coryphellidae sensu [22]. This situation is further complicated by complex mosaic character states within the entire diversity of Flabellinidae s.l., as initially identified by [19,23,24]. In many cases, key synapomorphies or shared features exhibit significant variability. For example, the presence of papillated rhinophores is an important synapomorphy of the genus *Coryphellina* (Fig 2), but within the genus *Coryphella* the morphology of the rhinophores varies substantially, from smooth to annulated, or perfoliated (Fig 2). Notably the high variability of rhinophore morphology was used to distinguish different genera within the Coryphellidae [22], but further phylogenetic studies did not support this suggestion, as rhinophores may differ significantly even in closely related species [34,41]. For example, *Coryphella fogata* has annulated rhinophores, and its sister species *C. cooperi* has wrinkled rhinophores [41], and the same was shown for *C. trophina* with perfoliate rhinophores and its closely related species *C. alexanderi* with smooth rhinophores [34,75].

In other cases, the variability of morphological characters requires considering their combination. This is exemplified by *Flabellina*. In the type genus of Flabellinidae, the arrangement of cerata, along with the features of the lateral teeth play a crucial role in identifying species within this genus. Except for *F. pellucida* and *F. enne*, all species in this genus are characterized by having cerata on compound peduncles (S3 Table), a feature unique to *Flabellina* within Flabellinidae (Fig 2). For the sister species *F. pellucida* and *F. enne*, the smooth lateral teeth (S3C, D Fig) set them apart from the genus *Edmundsella,* with which both species share several common traits: smooth rhinophores and cerata placed on low elevations (S2B, S15A–C Figs). The presence of these morphological characters in these latter two species indicates a retention of plesiomorphic traits.

Thus, despite the fact that the families Coryphellidae, Paracoryphellidae, and Flabellinidae sensu [22] were recovered as monophyletic and were highly supported in our BI analyses (Fig 1) (but not in the ML analysis, S1 Fig), the morphological study indicates rampant homoplasy across all groups. Further consideration of separate families Coryphellidae, Flabellinidae, and Paracoryphellidae with significantly overlapping diagnoses, as well as the description of many monotypic genera, leads to an unstable system that introduces practical difficulties associated with incorrect interpretation of characters' taxonomical value. Our results highlight the limitations of adopting a splitting approach in this context. For example, the species *Luisella telja* was placed in the genus *Samla* [22], and the species *Luisella engeli*, *Edmundsella dushia* and *E. bertschi* were considered members of the genus *Flabellina*, but [22] mentioned that these species may represent several separate genus-level taxa. On the one hand, our analyses confirm the affiliation of these species to the genera *Luisella* and *Edmundsella*, respectively, and also show that these groupings are supported by unique morphological synapomorphies characteristic of all species of the corresponding genera (Fig 2). In this regard, the re-definition of genus-ranking taxa to encompass larger species diversity provides significantly more practical advantages, since it eliminates the allocation of taxa based on single autapomorphies, and ultimately leads to a greater stability of the system and increases its predictability. This also eliminates the need to create a high number of monotypic genera for any new derived phylogenetic entity with unique autapomorphies. Therefore, we also propose to redefine the composition of Flabellinidae s.str. by synonymizing of the Coryphellidae and Paracoryphellidae with Flabellinidae.

We demonstrate that the traditional 'lumping' approach, grouping diverse taxa under the name *Flabellina*, obscures complex evolutionary patterns within Flabellinidae s.l. Our results generally support phylogenetic relationships recovered by [22], and we want to additionally emphasize that the referred work provided a robust baseline for phylogeny-based systematics. However, the instability of several newly added taxa highlights a weakness of their "splitting" approach, as excessive splitting produces narrow diagnoses thus decreasing predictability. Neither extreme lumping nor excessive splitting provides optimal solution. Consequently, a balanced approach provides more advantages to accommodate mosaic evolution, overlapping diagnoses, and enhance practical stability. This balance aims for a more resilient classification.

A recent phylogenetic reconstruction and taxonomic revision of Aeolidacea [35] broadly agrees with our results in terms of recovered clades, but differs substantially in translating phylogeny into taxonomy. Most notably, the clade termed Flabellinidae by [35] is not supported at a reasonable level in any analysis, whereas the clade we recognize as Flabellinidae here is consistently recovered, well supported, and assigned family rank. Similarly, the grouping of *Flabellina*, *Paraflabellina*, *Calmella*, and *Carronella* at a level of genetic divergence comparable to *Coryphellina* and *Edmundsella* is supported in both studies despite [22,35] using a smaller dataset. The main taxonomic discrepancies stem from differing approaches to defining taxa: [22,35] emphasizes splitting based on interpreted morphological differences, which risks overlooking inconsistencies and relies on inherently subjective interpretation. While integrating phylogenetic and morphological data can reveal finer evolutionary patterns, especially in well-studied groups, clades corresponding to plausible taxa can be identified in most phylogenies, and supporting arguments can usually be constructed post hoc. A more robust approach should actively test and question putative taxa, considering clade stability, support, and the limitations of morphological diagnoses. This is especially important in groups with extensive undescribed diversity, where premature

high-level taxonomic revisions may destabilize classification. Given that new specimens frequently prompt reassessment of both phylogenetic relationships and morphological interpretations, current datasets and analyses remain incomplete, and early studies often underestimate true diversity. Consequently, major taxonomic changes based on [35] approach risk rapid obsolescence. In line with a Bayesian perspective, we argue that taxonomy should be revised only when successive analyses of expanded datasets consistently support proposed taxa and reveal no evidence against them.

## Conclusion

In conclusion, the results of our study, along with our interpretation of the data, suggest that a more conservative approach to the systematics of Flabellinidae s.l. offers multiple advantages, including more simplicity, better defined taxa, and congruence with molecular phylogenetic data. This approach is a clear departure from more complex schemes recently proposed for Flabellinidae s.l., but it is still a much more intricate system compared to simpler traditional classifications of this group. Molecular data have clearly helped to alleviate issues derived from the rampant homoplasy in Flabellinidae s.l., but the subjective interpretation of taxonomic categories above species (e.g., genera, families) remains the holy grail of modern systematics. In the case of Flabellinidae s.l., this matter is clearly not settled, numerous undescribed species still exist, and their discovery may result in further changes to the classification. Moreover, the lack of data on some critical taxa (e.g., the type species of *Piseinotecus*) remains a serious problem to resolve synonymies at the family level. The advent of genomics will help to bring more reliable phylogenies, less dependent on the pervasive effects of limited taxonomic coverage, but will not resolve the fundamental issue at hand: the subjectivity of interpretation of taxonomic categories above species. Thus, the everlasting tension between lumpers and splitters rages on.

## Supporting information

**S1 Data. Unedited single-gene trees (COI, 16S, H3, 28S) from ML and unedited concatenated trees from ML and BI.**
(ZIP)

**S1 Table. Specimens used in the present study with information on vouchers and GB accession numbers.**
(XLSX)

**S2 Table. Primers used for amplification and sequencing and respective amplification programs.**
(DOCX)

**S3 Table. Morphological data for all currently accepted Flabellinidae, Flabellinopsidae, Samlidae, Apataidae, Piseinotecidae, Notaeolididae species.**
(XLSX)

**S4 Table. P-distances matrix calculated for all Flabellinidae species studied herein.**
(XLSX)

**S5 Table. P-distances matrix between species of *Launsina* gen. nov. and *Mgueolia.***
(XLSX)

**S1 Fig. Molecular phylogenetic hypothesis based on the Maximum likelihood (ML), concatenated dataset of four markers (COI+16S+H3+28S), outgroups are collapsed to a single branch. Species names on leaves are given according to Korshunova et al. (2017) and Ekimova et al. (2024).** Numbers on nodes indicate bootstrap support from ML.
(TIF)

**S2 Fig. The genus *Flabellina,* external morphology of different species (vouchers not specified).** A – *Flabellina affinis.* B – *Flabellina pellucida.* C – *Flabellina ischitana.* D – *Flabellina gaditana.* E, F – *Flabellina cavolini.* Photo credits: A, C – Manuel Martínez Chacón. B – Tatiana Antokhina. D – Sául Patiño Gómez. E, F – Enric Madrenas.
(TIF)

**S3 Fig. Radular morphology in genus *Flabellina* (SEM).** A, B – *Flabellina affinis,* MNCN15.05/98993. C, D – *Flabellina pellucida,* IEIn1 (C) and IEFp8 (D). E, F – *Flabellina cavolini,* MNCN15.05/98991. G, H – *Flabellina ischitana*, MNCN15.05/98990. Scale bars: A–E, G = 10 μm. F = 5 μm. H = 3 μm.
(TIF)

**S4 Fig. Morphology of jaws in genus *Flabellina* (SEM).** A, B – *Flabellina ischitana,* MNCN15.05/98990, jaw body (A) and masticatory border (B). C, D – *Flabellina cavolini,* MNCN15.05/98991, jaw body (C) and masticatory border (D). Scale bars: A, C = 100 μm. B, D = 10 μm.
(TIF)

**S5 Fig. Major patterns of the reproductive system configuration in the genus *Flabellina.*** A – *Flabellina affinis.* B – *Flabellina cavolini.* C – *Flabellina funeka.*
(TIF)

**S6 Fig. The genus *Coryphella,* external morphology of different species.** A – *Coryphella verrucosa* ZMMU WS14967. B – *Coryphella sanamyanae* ZMMU WS14987. C – *C. nobilis,* voucher not specified. D – *C. amabilis* ZMMU WS14957. E – *Coryphella orjani* ZMBN130734. F – *C. chriskaugei* ZMBN125918. G – *C. lineata* ZMBN127566. H – *C. browni* ZMBN126001. I – *Coryphella* sp. 2 CPIC1639. J – *C. trilineata* CAS218410. K – *C. cooperi* CPIC1646. L – *C. athadona,* ZMMU WS14999. Photo credits: A, B, D, L – Tatiana Antokhina, Yury Deart. C – Sergey Gorin. E–H – Manuel Malaquias. I – Ángel Valdés. J – Brenna Green. K – Craig Hoover.
(TIF)

**S7 Fig. Radular morphology in genus *Coryphella* (SEM).** A – *Coryphella verrucosa,* ZMMU WS14451. B – *Coryphella nobilis,* ZMMU WS14379. C – *Coryphella sanamyanae,* ZMMU WS14416. D – *Coryphella alexanderi,* MIMB42469. E – *Coryphella trophina*, ZMMU WS14405. F – *Coryphella gracilis,* ZMMU WS14915. G – *Coryphella falklandica,* ZIN N9. H – *Coryphella* sp. 2, CPIC880. I – *Coryphella cooperi,* CPIC1646. J – *Coryphella athadona,* ZMMU WS14424. Scale bars: A–F, H, I = 30 μm. G = 50 μm. J = 20 μm.
(TIF)

**S8 Fig. Morphology of masticatory border of jaws in genus *Coryphella* (SEM).** A – *Coryphella verrucosa,* ZMMU WS14440. B – *Coryphella nobilis,* MIMB40023. C – *Coryphella sanamyanae,* ZMMU WS14417. D – *Coryphella trophina,* ZMMU WS14435. E – *Coryphella gracilis,* ZMMU WS14914. F – *Coryphella falklandica,* ZIN N9. G – *Coryphella alexanderi,* MIMB42468. H – *Coryphella* sp. 2, CPIC880. I – *Coryphella cooperi,* CPIC1646. Scale bars: A, F, I = 50 μm. B, C, E, G, H = 30 μm. D = 100 μm.
(TIF)

**S9 Fig. Major patterns of the reproductive system configuration in the genus *Coryphella.*** A – *Coryphella falklandica.* B – *Coryphella gracilis.* C – *Coryphella verrucosa.* D – *Coryphella verta.*
(TIF)

**S10 Fig. Representatives of the genus *Chlamylla*: external morphology (upper line), jaw and radular morphology (SEM).** A – *Chlamylla intermedia,* BBS23017. B – *Chlamylla borealis orientalis,* Sh-1. C – *Chlamylla islandica,* BBS24116.

D – *Chlamylla intermedia*, voucher not specified, radula. E – *Chlamylla islandica,* BBS24116, radula. F – *Chlamylla* cf. *polaris,* abyssal depth of Sea of Japan, MIMB49484, radula. G, H – *Chlamylla intermedia,* voucher not specified, radula. Photo credits: A, C – Irina Ekimova; B – Andrey Shpatak. Scale bars: D–F, H = 20 μm. G = 200 μm.
(TIF)

**S11 Fig. Major patterns of the reproductive system configuration in the genera *Chlamylla* and *Ziminella.* A –** *Chlamylla intermedia.* B – *Chlamylla islandica.* C – *Chlamylla polaris.* D – *Ziminella vrijenhoeki.*
(TIF)

**S12 Fig. The genus *Coryphellina,* external morphology of different species. A** – *Coryphellina rubrolineata,* voucher not specified. B – *Coryphellina lotos,* IZ-T9. C – *Coryphellina pseudolotos* N70. D – *Coryphellina pannae* ZIN63217. E – *Coryphellina flamma* ZIN63218. F – *Coryphellina* sp. 3 N64. G – *Coryphellina* sp. 2 QUP14_AC204. H – *Coryphellina flamma* QUP14_AC179. I – *Coryphellina aurora,* ZIN63224. J – *Coryphellina* sp. 1 N4. K – *Coryphellina exoptata* QUP56_AC570. Photo credits: B – Elena Mekhova, C–F, I, J –Yury Deart, Tatiana Antokhina. G, H, K – Ángel Valdés.
(TIF)

**S13 Fig. Radular (upper rows) and masticatory border (lower row) morphology in genus *Coryphellina* (SEM).**
A – *Coryphellina rubrolineata,* IE-fr4. B – *Coryphellina aurora,* ZIN63224. C – *Coryphellina lotos,* IZ-L4. D – *Coryphellina pseudolotos,* N67. E – *Coryphellina flamma*, N63. F – *Coryphellina* sp. 1, L115. G – *Coryphellina rubrolineata*, IE-fr2. H – *Coryphellina aurora*, ZIN63224. I – *Coryphellina flamma,* ZIN63219. Scale bars: A, G = 30 μm. B, C, I = 50 μm. D, F = 10 μm. E, H = 20 μm.
(TIF)

**S14 Fig. Major patterns of the reproductive system configuration in the genera *Coryphellina* and *Edmundsella.* C** – *Coryphellina albomarginata.* D – *Coryphellina rubrolineata.* C – *Edmundsella bertschi.* D – *Edmundsella dushia.*
(TIF)

**S15 Fig. Representatives of the genus *Edmundsella*: external morphology (upper line). jaw and radular morphology (SEM). A** – *Edmundsella pedata,* Bahia de Algeciras, voucher not specified. B – *Edmundsella bertschi,* CPIC1648. C – *Edmundsella dushia,* voucher not specified. D – *Edmundsella* sp., CPIC843, right jaw plate. E – *Edmundsella pedata,* M1, anterior radular portion. F – *Edmundsella bertschi,* CPIC1648, anterior radular portion. G – *Edmundsella* sp., CPIC843, anterior radular portion. Photo credits: A – Manuel Martínez Chacón; B, C – Ángel Valdés. Scale bars: D = 100 μm. E, F = 10 μm. G = 30 μm.
(TIF)

**S16 Fig. Representatives of the genus *Ziminella*: external morphology (left line), jaw and radular morphology (SEM). A** – *Ziminella vrijenhoeki,* MIMB42255. B – *Ziminella vrijenhoeki,* MIMB42255, radula. C – *Ziminella vrijenhoeki,* MIMB42255, masticatory border of jaw. D – *Ziminella japonica*, MIMB50760. E – *Ziminella japonica*, MIMB50759−1, radula. F – *Ziminella japonica*, MIMB50759−1, masticatory border of jaw. Photo credits: A – Anastasia Maiorova; D – Olga Chichvarkhina. Scale bars: D–F, H = 20 μm. G = 200 μm.
(TIF)

**S17 Fig. *Flabellinopsis iodinea* MacFarland, 1966, external and internal morphology. A** – voucher not specified, specimen from Palos Verdes, Southern California, USA. B – CPIC1015. C – voucher not specified, specimen from Bahía de los Ángeles, Baja California, México. D – CPIC1459, left jaw plate. E – CPIC1459, details of masticatory border denticulation. F – CPIC1459, median and posterior radular portions. G – CPIC1033, rachidian and lateral teeth, median radular portion. H – CPIC1459, lateral tooth, note denticulation of outer surface. Scale bars: D = 100 μm. F, E, H = 10 μm. G = 50 μm.
(TIF)

**S18 Fig. Major patterns of the reproductive system configuration in the family Flabellinopsidae.**
(TIF)

**S19 Fig.** *Baenopsis baetica* **(García-Gómez, 1984), external and internal morphology.** A – living photo, specimen was not collected. B–G – MNCN15.05/98989, buccal armature (SEM). B – masticatory border of jaw. C – radula. D – rachidian teeth. E – lateral tooth. F, G – details of lateral teeth outer (F) and inner (G) denticulation. Scale bars: B = 50 μm. C = 100 μm. D = 10 μm. E = 30 μm. F, G = 5 μm.
(TIF)

**S20 Fig.** *Kynaria cynara* **(Ev. Marcus & Er. Marcus, 1967), external and internal morphology.** A – CPIC579. B–D – CPIC581. E – right jaw plate, CPIC579. F – radula, CPIC579. G – rachidian and lateral teeth, CPIC579. H – details of denticulation of masticatory process. I – details of denticulation of lateral teeth outer surface. Scale bars: E, F = 300 μm. G = 100 μm. H = 30 μm. I = 10 μm.
(TIF)

**S21 Fig.** *Bajaeolis bertschi* **Gosliner & Behrens, 1986, external and internal morphology.** A – CPIC0597. B – CPIC0582. C – CPIC0592. D – paired jaw plates, CPIC592. E – radula, CPIC592. F – rachidian teeth, posterior radular portion, CPIC592. G – rachidian teeth, middle radular portion, CPIC592. H – details of denticulation of jaws masticatory border. Scale bars: D = 100 μm. E–H = 20 μm.
(TIF)

**S22 Fig. Representatives of the genus** *Samla*: **external morphology (upper line), jaw and radular morphology (SEM).** A – *Samla bicolor,* n19. B – *Samla takashigei,* n378. C – *Samla* sp. 1, n20. D – *Samla* sp. 2, n325. E – *Samla riwo,* QNAL-KA-GF005. F – *Samla takashigei,* n378, radula. G – *Samla bicolor,* n52, radula. H – *Samla riwo,* QNAL-HA-JD346, radula. I – *Samla bicolor,* n52, jaw. J – *Samla bicolor,* n52, masticatory border of jaw. K – *Samla takashigei,* n378, masticatory border of jaw. Photo credits: A–D – Yury Deart, Tatiana Antokhina. F – Angel Valdes. Scale bars: F–H = 20 μm. I = 200 μm. J, K = 10 μm.
(TIF)

**S23 Fig. Major patterns of the reproductive system configuration in the family Samlidae.** A – *Samla bicolor.* B – *Luisella engeli.* C – *Luisella babai.* D – *Luisella telja.*
(TIF)

**S24 Fig. Representatives of the genus** *Luisella*: **external morphology, jaw and radular morphology (SEM).** A–E – *Luisella engeli,* CPIC1582. F–H – *Luisella telja,* CPIC2110. A – living specimen. B – anterior part of jaw. C – masticatory process of jaw. D – *radula.* E – posterior radular portion. F – living specimen. G – radula. H – posterior radular portion. I – masticatory process of jaw. K – rachidian tooth. Scale bars: B, D = 100 μm. C, K = 10 μm. E, H, I = 30 μm. G = 300 μm.
(TIF)

**S25 Fig. Representatives of the genus** *Apata*: **external morphology, jaw and radular morphology (SEM).** A – *Apata* cf. *pricei*, from Bahía de los Ángeles, Mexico, not collected. B, C – *Apata* cf. *pricei*, from San Diego, California, not collected. D, E – *Apata pricei kommandorica,* voucher not specified. F, G, J, K – Rud21177. H, I, L – Rud21284. M – CPIC1020. F, H – jaw. G, I – masticatory process. J–M – radula. A – living specimen, voucher not specified. B – anterior part of jaw. C – masticatory process of jaw. D – radula. E – posterior radular portion. F – living specimen. G – radula. H – posterior radular portion. I – masticatory process of jaw. K – rachidian tooth. Photo credits: A–C: Craig Hoover. D, E: Andrey Shpatak. Scale bars: F = 300 μm. G, K, M = 30 μm. H, J = 100 μm. I, L = 50 μm.
(TIF)

**S26 Fig. Major patterns of the reproductive system configuration in the families Apataidae and Notaeolidiidae.**
(TIF)

**S27 Fig. Representatives of the genus *Notaeolidia*: external morphology, jaw and radular morphology.** A – *Notaeolidia* cf. *depressa,* Nd1. B–E – *Notaeolidia depressa,* ZIN1. B, C – radula. D, E – jaws. F – *Notaeolidia* cf. *depressa,* Nd1, radula. Photo credit: A – Olga Bozhenova. Scale bars: B = 100 µm. C, F = 50 µm. D = 10 µm. E = 500 µm.
(TIF)

**S28 Fig. Major patterns of the reproductive system configuration in the family Piseinotecidae.** A – *Piseinotecus divae.* B – *Piseinotecus soussi.* C – *Piseinotecus minipapilla*. D – *Piseinotecus gonja*. E – *Unidentia angelvaldesi*. F – *Pacifia goddardi.*
(TIF)

**S29 Fig. Representatives of the genus *Unidentia*: external morphology, jaw and radular morphology (SEM).** A – *Unidentia aliciae,* N101. B – *Unidentia sandramillenae,* N432. C – *Unidentia* sp. 1, N313. D, E – *Unidentia aliciae,* N102, radula. F – *Unidentia sandramillenae,* N432, radula. G, H – *Unidentia aliciae,* N101, jaws. Photo credits: A–C – Yury Deart, Tatiana Antokhina. Scale bars: D, F = 30 µm. E = 10 µm. G = 100 µm. H = 50 µm.
(TIF)

**S30 Fig. Molecular phylogenetic hypothesis based on the Maximum likelihood (ML), testing the position of *Edmundsella dushia* (BOLD voucher MUSBA1086−24), concatenated dataset of four markers (COI + 16S + H3 + 28S), including only representatives of Flabellindiae s.l., and Aeolidiidae + Tritoniidae as outgroups.** Numbers on nodes indicate bootstrap support from ML.
(TIF)

## Acknowledgements

The expedition to New Caledonia operated under permits no. 609002–31/2018/DEPART/YM and 609011–55/2019/DEPART/JJC were issued respectively 27 August 2018 and 22 October 2019 by Direction du Développement Economique et de l'Environnement (DDEE) of Province Nord. We are grateful to Philippe Bouchet for the invitation to participate in these expeditions.

We are indebted to the A.V. Zhirmunsky National Scientific Center of Marine Biology, Far Eastern Branch of Russian Academy of Sciences (NSCMB FEB RAS), for organizing and conducting the 73rd cruise of the RV Akademik Oparin. We thank the captain, crew, and scientific team for their support during the expedition.

We are deeply grateful to all people who kindly collected and donated material for this study: Tatiana Antokhina, Yury Deart, Enric Madrenas, Saúl Patiño Gómez, Marta Pola, Joaquim Reis, Juan Lucas Cervera, Olga Bozhenova, Anastasia Mayorova, Olga Chichvarkhina, Andrey Shpatak, Katia Nakamura, Ari Dimitris, Glafira Kolbasova, Elena Vortsepneva. We are deeply thankful to all individuals who provided images to this study: Enric Madrenas, Saúl Patiño Gómez, Manuel Martínez Chacón. Thanks to Ana Karla Araújo from the University of Cadiz for her help with radula preparation. We also want to thank Valentina Tambovtseva and Daria Rozhkova from Institute of Developmental Biology, Russian Academy of Sciences for assistance in Sanger sequencing. The light microscopy (partly) and molecular studies (partly) were conducted using equipment of the Invertebrate zoology Department MSU, the electron microscopy studies (partly) — using equipment the Shared Research Facility "Electron microscopy in life sciences" at Moscow State University (Unique Equipment "Three-dimensional electron microscopy and spectroscopy") and equipment of the Center of microscopy N.A. Pertsov White Sea Biological Station MSU. Sanger sequencing (partly) was conducted using equipment of the Core Centrum of Institute of Developmental Biology RAS and Invertebrate zoology department, Lomonosov Moscow State University.

## Author contributions

**Conceptualization:** Irina Ekimova, Leila Carmona, Ángel Valdés.

**Data curation:** Irina Ekimova, Leila Carmona, Ángel Valdés.

**Formal analysis:** Irina Ekimova, Leila Carmona, Anna L. Mikhlina, Darya Grishina, Maria V. Stanovova, Craig Hoover, Jade de Souza-Canal, Karen O. Kuznetsov.

**Funding acquisition:** Irina Ekimova, Leila Carmona, Ángel Valdés.

**Investigation:** Irina Ekimova, Leila Carmona, Anna L. Mikhlina, Darya Grishina, Maria V. Stanovova, Dimitry M. Schepetov, Jade de Souza-Canal, Karen O. Kuznetsov.

**Methodology:** Irina Ekimova, Leila Carmona, Dimitry M. Schepetov.

**Project administration:** Irina Ekimova.

**Resources:** Irina Ekimova, Leila Carmona, Anna L. Mikhlina, Darya Grishina, Maria V. Stanovova, Craig Hoover, Jade de Souza-Canal, Karen O. Kuznetsov, Ángel Valdés.

**Software:** Irina Ekimova, Dimitry M. Schepetov.

**Supervision:** Irina Ekimova, Leila Carmona, Ángel Valdés.

**Validation:** Irina Ekimova, Leila Carmona, Ángel Valdés.

**Visualization:** Irina Ekimova, Anna L. Mikhlina, Darya Grishina.

**Writing – original draft:** Irina Ekimova, Leila Carmona, Ángel Valdés.

**Writing – review & editing:** Irina Ekimova, Leila Carmona, Anna L. Mikhlina, Darya Grishina, Maria V. Stanovova, Dimitry M. Schepetov, Craig Hoover, Jade de Souza-Canal, Karen O. Kuznetsov, Ángel Valdés.

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
