## [Decision Letter · Decision Letter 0]

6 Nov 2025

PONE-D-25-43242Neither “lumpers” nor “splitters”: A global revision of Flabellinidae s.l. nudibranchs (Gastropoda: Heterobranchia: Nudibranchia)PLOS ONE

Dear Dr. Ekimova,

Thank you for submitting your manuscript to PLOS ONE. After careful consideration, we feel that it has merit but does not fully meet PLOS ONE’s publication criteria as it currently stands. Therefore, we invite you to submit a revised version of the manuscript that addresses the points raised during the review process.

If applicable, we recommend that you deposit your laboratory protocols in protocols.io to enhance the reproducibility of your results. Protocols.io assigns your protocol its own identifier (DOI) so that it can be cited independently in the future. For instructions see: https://journals.plos.org/plosone/s/submission-guidelines#loc-laboratory-protocols. Additionally, PLOS ONE offers an option for publishing peer-reviewed Lab Protocol articles, which describe protocols hosted on protocols.io. Read more information on sharing protocols at . Additionally, PLOS ONE offers an option for publishing peer-reviewed Lab Protocol articles, which describe protocols hosted on protocols.io. Read more information on sharing protocols at https://plos.org/protocols?utm_medium=editorial-email&utm_source=authorletters&utm_campaign=protocols..

We look forward to receiving your revised manuscript.

Kind regards,

Michael Schubert

Academic Editor

PLOS ONE

Journal Requirements:

2. Please take this opportunity to be sure you have met all of our guidelines for new species. For proper registration of a new zoological taxon, we require two specific statements to be included in your manuscript.

In the Results section, the globally unique identifier (GUID), currently in the form of a Life Science Identifier (LSID), should be listed under the new species name, for example:

Anochetus boltoni Fisher sp. nov. urn:lsid:zoobank.org:act:B6C072CF-1CA6-40C7-8396-534E91EF7FBB

Another LSID for the manuscript itself should also appear within the Nomenclature statement. You will need to contact Zoobank (zoobank.org/About) to obtain a GUID (LSID). You should receive one LSID for your manuscript and a separate, unique LSID for the new species.

Please also insert the following text into the Methods section, in a sub-section to be called "Nomenclatural Acts":

The electronic edition of this article conforms to the requirements of the amended International Code of Zoological Nomenclature, and hence the new names contained herein are available under that Code from the electronic edition of this article. This published work and the nomenclatural acts it contains have been registered in ZooBank, the online registration system for the ICZN. The ZooBank LSIDs (Life Science Identifiers) can be resolved and the associated information viewed through any standard web browser by appending the LSID to the prefix "http://zoobank.org/". The LSID for this publication is: urn:lsid:zoobank.org:pub: XXXXXXX. The electronic edition of this work was published in a journal with an ISSN, and has been archived and is available from the following digital repositories: PubMed Central, LOCKSS [author to insert any additional repositories].

All PLOS ONE articles are deposited in PubMed Central and LOCKSS. If your institute, or those of your co-authors, has its own repository, we recommend that you also deposit the published online article there and include the name in your article.

Following a recent ruling by the International Commission on Zoological Nomenclature, electronic journals are now a valid format for publication of new zoological taxa. In order to ensure the valid publication of your new species, please be sure to include the updated version of Nomenclatural Acts (above). A complete explanation of our guidelines for publishing new species can be found on our website: http://www.plosone.org/static/guidelines#zoological

Reviewers' comments:

Reviewer's Responses to Questions

**Comments to the Author**

1. Is the manuscript technically sound, and do the data support the conclusions?

Reviewer #1: Yes

Reviewer #2: Yes

Reviewer #3: Yes

2. Has the statistical analysis been performed appropriately and rigorously? 

Reviewer #1: Yes

Reviewer #2: N/A

Reviewer #3: Yes

3. Have the authors made all data underlying the findings in their manuscript fully available?

Reviewer #1: Yes

Reviewer #2: Yes

Reviewer #3: Yes

4. Is the manuscript presented in an intelligible fashion and written in standard English?

Reviewer #1: Yes

Reviewer #2: Yes

Reviewer #3: Yes

5. Review Comments to the Author

Reviewer #1: The authors present a complex and detailed study on the systematics of the family Flabellinidae, a group whose taxonomic status has long been problematic due to both historical and recent conflicting classifications. The manuscript provides a useful and comprehensive overview of the history of classification issues in the family which serves as a solid framework for the study. The amount of data analyzed and the methodology employed are appropriate, reflecting meticulous work and considerable effort that deserves acknowledgment. The integration of phylogenetic analyses with morphological traits is a major strength of the manuscript.

The paper is generally well written and logically structured. That said, the text would benefit from a light language revision to shorten some of the longer sentences and to reduce repetition, making it more concise and easier to follow. This would improve readability and enhance the impact of the results on the reader. I am attaching a PDF with some specific comments line by line.

Overall, the authors demonstrate strong command of methods and background knowledge. The study represents a valuable contribution that advances our current understanding of flabellinid systematics. The high-quality figures and graphics further support the work. In addition the supplementary material is very valuable. All this is clearly the result of substantial effort, which should be recognized and valued.

Reviewer #2: The research presented in the MS PONE-D-25-43242 entitled Neither “lumpers” nor “splitters”: A global revision of Flabellinidae s.l. nudibranchs (Gastropoda: Heterobranchia: Nudibranchia) by Ekimova et al., is proposing a new taxonomical scheme for Flabellinidae s.l., a group of nudibranchs overlong studied in the past but still difficult to interpret considering morphological characters and molecular markers.

In the frame of the work, the old lasting discussion about the conflict between “lumpers and “splitters” is commented within the critical revision of the Flabellinidae s.l. The final result obtained is a new taxonomical scheme for Flabellinidae s.l., integrating traditional morphology and molecular DNA analyses, using four standard molecular markers. This MS is without any doubt very interesting and has been produced from a huge amount of work. The MS is worthy of publication, since it is a novel and important contribution to understanding the systematic of the Flabellinidae s.l. and well adheres to the main points requested by the policy of PONE. I strongly recommend publishing of this MS in the present form. I only suggest mentioning in Discussion also the discrepancy between the phylogenetics reconstructions obtained by the recent work of Korshunova et al. [Zoological Journal of the Linnean Society, 2025, 204(4)] versus the one presented in this MS.

Reviewer #3: Review of Manuscript PONE-D-25-43242: “Neither “lumpers” nor “splitters”: A global revision of Flabellinidae s.l. nudibranchs (Gastropoda: Heterobranchia: Nudibranchia)”

This manuscript by Ekimova et al revisits the systematics of Flabellinidae sensu lato to investigate former classifications in light of new data and discuss challenges in nudibranch systematics more broadly that work in this group has brought up. I find this manuscript thoughtful and thorough, with clear reasoning for nomenclatural and taxonomic decisions. Although the phylogeny is still in flux at the deeper nodes, the vast majority of named clades are supported as monophyletic. Where sufficient uncertainty exists, the authors do a good job exerting caution with regard to changing names. This paper also describes new genera and species that make sense in the context of the molecular and morphological data. My primary suggestions are minor, see below.

Methods:

Lines 162-163: Were further partitioning models investigated (i.e., by codon position)? It would be useful to explore these models at the very least.

Results:

Overall: I appreciate why the authors structured the first part of the results section by the previous classification, but it is confusing considering that some of these taxa do not exist in their proposed reclassification (see Coryphellidae). Perhaps a more explicit note that they are addressing the prior classification would be helpful.

Overall: Please provide specific references to figures in the descriptions (e.g., Figure S2A) to make it easier to connect the figures to the text.

Overall: Given that Figure S1 is so different, many of these genera/families should be compared with other taxa more broadly, since their affinities are still so uncertain.

Line 257: Flabellina sp. is listed as Edmunsella sp. 1 in the tree as far as I can tell.

Line 315: How much genetic divergence?

Lines 332-435: It is confusing that these are not listed in the same order as in Figure 2. Consistency would make this section much easier to follow when looking at the figure.

Line 472: In the drawing, the distal seminar receptacle does not look bilobed. It looks like there is a second one.

Line 503: Have smooth rhinophores and non-pedunculate cerata been characterized as plesiomorphic states? This should either be rephrased, or some sort of ancestral state reconstruction should be done to provide some evidence for this claim.

Line 528: I don’t see a distal seminal receptacle in Fig. S9A

Lines 579-580: “narrow to wide triangular…” I’m assuming refers to the lateral teeth?

Lines 587-589: Please provide some context for why these two species are included in Chlamylla. What morphological characters are being used to decide? This is something I noticed across other descriptions as well.

Line 620: I wouldn’t call these bilobed. Based on the drawings, these appear to be two separate proximal seminal receptacles.

Line 655 (and others): I noticed multiple times that the authors use “vas deference” rather than “vas deferens”

Lines 669-670: Some similarities with Coryphellina as well, though that is not mentioned.

Lines 727-728: Is there a reason this is not mentioned in the diagnosis?

Line 784: for should be from

Lines 880-881: The location of the vagina is not clear in Fig 4C

Line 972: centrally should be proximally

Line 975: Figure 6D should be 4D

Line 1001: Bajaeolis is listed as uniserial in Figure 2

Line 1002: Figure 2 shows only uncompressed

Line 1028: Only proximal in S18B, I also don’t see the wide prostate in the figure

Line 1043: Don’t see the wide prostate in the figure

Line 1053-1054: How is the radula triserial if the lateral teeth are absent?

Lines 1138-1151: What species are included in the genus Apata? Should provide such information for consistency.

Line 1171: With ridges? See Figure 2

Line 1172: “triangular or elongated…” Is this referring to the rachidian teeth?

Line 1175: Proximal seminal receptacle not shown in Figure S26C

Line 1190: placed on low elevations?

Line 1229: The drawings in Figure S28 don’t show a wide, distinct prostate

Discussion:

Line 1422: annulled should be annulated I assume

Figures:

Please provide all tree files in the supplement, including for the concatenated data sets.

6. PLOS authors have the option to publish the peer review history of their article (what does this mean?). If published, this will include your full peer review and any attached files.). If published, this will include your full peer review and any attached files.

.

Reviewer #1: No

Reviewer #2: No

Reviewer #3: No

---

## [Author Response · Author response to Decision Letter 1]

26 Feb 2026

Response to reviewers, manuscript “Neither “lumpers” nor “splitters”: A global revision of Flabellinidae s.l. nudibranchs (Gastropoda: Heterobranchia: Nudibranchia)” by Ekimova et al.

Reviewer #1: The authors present a complex and detailed study on the systematics of the family Flabellinidae, a group whose taxonomic status has long been problematic due to both historical and recent conflicting classifications. The manuscript provides a useful and comprehensive overview of the history of classification issues in the family which serves as a solid framework for the study. The amount of data analyzed and the methodology employed are appropriate, reflecting meticulous work and considerable effort that deserves acknowledgment. The integration of phylogenetic analyses with morphological traits is a major strength of the manuscript.

The paper is generally well written and logically structured. That said, the text would benefit from a light language revision to shorten some of the longer sentences and to reduce repetition, making it more concise and easier to follow. This would improve readability and enhance the impact of the results on the reader. I am attaching a PDF with some specific comments line by line.

Thank you very much for a high evaluation, we have accepted all this suggestions and modified the text accordingly.

Overall, the authors demonstrate strong command of methods and background knowledge. The study represents a valuable contribution that advances our current understanding of flabellinid systematics. The high-quality figures and graphics further support the work. In addition the supplementary material is very valuable. All this is clearly the result of substantial effort, which should be recognized and valued.

General comments

The authors present a complex and detailed study on the systematics of the family Flabellinidae, a group whose taxonomic status has long been problematic due to both historical and recent conflicting classifications. The manuscript provides a useful and comprehensive overview of the history of classification issues in the family which serves as a solid framework for the study. The amount of data analyzed and the methodology employed are appropriate, reflecting meticulous work and considerable effort that deserves acknowledgment. The integration of phylogenetic analyses with morphological traits is a major strength of the manuscript.

The paper is generally well written and logically structured. That said, the text would benefit from a light language revision to shorten some of the longer sentences and to reduce repetition, making it more concise and easier to follow. This would improve readability and enhance the impact of the results on the reader.

Overall, the authors demonstrate strong command of methods and background knowledge. The study represents a valuable contribution that advances our current understanding of flabellinid systematics. The high-quality figures and graphics further support the work. This is clearly the result of substantial effort, which should be recognized and valued. I recommend the manuscript for publication after minor revisions.

Abstract

General comments: While the abstract is clear and it successfully conveys the objectives and rationale of the study it is a bit too long and it would benefit from some tightening and rephrasing to improve clarity and impact. The key results and taxonomic proposals are important, but too much detail is included, please consider summarizing the new taxa and emphasizing the broader systematic implications. A short abstract is always “powerful”.

Thank you! Abstract was reduced following these suggestions

Line 29-30: The sentence beginning with “it is extremely challenging...” could be separated into two sentences.

corrected

Line 31: The phrase “in its traditional sense” is vague. I also believe that the sentence could be rephrased to avoid repetition (e.g.: “Recent studies have shown that the group is polyphyletic and have suggested reclassifying the 84 described species into 29 genera across 7 families.”)

corrected

Line 33: I´d eliminate “on the taxonomy and phylogeny”, as this is already implied.

corrected

Line 34: Replace “and/or” with just “and”.

corrected

Line 44: Typo: “results shown” should be “results showed” or “results show”, depending on whether the authors want to keep the abstract in past or present tense.

corrected

Line 52: Replace “most likely represents“ with just “likely represents”.

corrected

Introduction

General comments: A useful and comprehensive overview of the long history of classification issues in Flabellinidae s.l. is provided. However, it is a bit overly detailed, which risks losing the reader. I recommend reducing background information that has already been summarized in earlier reviews (e.g., [22]) and focusing more directly on the research gap to justify the present study. Some sentences are quite long and can be modified to improve readability.

corrected

Line 58: Rephrase to (…) conflict between ‘lumpers’ and ‘splitters [1]. This issue, mainly driven by reliance on ambiguous morphological traits… (change position ambiguous to improve gramma).

corrected

Line 59–60: Simplify to “…resulted in both over- and underestimations of diversity. The existence of cryptic diversity further exacerbated the underestimation problem…”

corrected

Line 65: “suggesting and end” instead of “evoking the end” is in my opinion a better choice of words.

corrected

Line 68: I recommend adding “can” so the sentence reads: “where the lack of clear definitions can make taxonomic decisions on genera and families highly subjective.” This small change emphasizes that subjectivity is a potential issue rather than a common situation. In my opinion, this nuance is important, as taxonomy is not inherently subjective when supported by thorough and well-structured work.

corrected

Line 69–71: Break into two sentences (as my suggestion in the abstract) “…cryptic diversity, plastic ecology, and variable COI divergence across groups make morphological interpretation extremely challenging. As a result, it is difficult to reach a consensus…”

corrected

Line 73: I suggest simplifying the sentence (e.g.: Flabellinidae s.l. is a large cosmopolitan group of nudibranchs, with more than 90 described species [8])

corrected

Line 75–77: This sentence can be break after “two ‘main’ genera”

corrected

Line 81–83: “in contrast” can be replaced by “vs” to simplify

corrected

Line 111–113: Simplify to “…Most of these genera” instead of “Most of the genus-level taxa”

corrected

Material and Methods

General comments: This section is detailed and covers all the needed information. However, some sentences can be modified a bit to make them more concise or grammatically precise.

corrected

Line 132: Rephrase: “For this study, we used specimens from extensive museum collections deposited at…”

corrected

Lines 133–136: The list of institutions can be a bit clearer by adding some information between brackets: e.g., California State Polytechnic University Invertebrate Collection (CPIC, Pomona, USA), (MNHN, Paris, France)

corrected

Line 138: Modify to “between 2011 and 2024.”

corrected

Line 143: It should be “…are described in detail in [34–36].” (detail without the final S)

corrected

Line 144: The use of dashes makes the reading a bit difficult, instead you could write it more smoothly ““…described in detail in [34–36]; those used at California State Polytechnic University are described in [37]; and those at the Instituto Universitario de Investigación Marina (INMAR), University of Cádiz, in [38].”

corrected

Line 153–154: This sentence can be reduced (e.g.:“Previously published datasets [22, 28, 34, 40] were incorporated, with Tritoniidae selected as a distant outgroup along with representatives of all major families of Aeolidoidea.”)

corrected

Line 171: Consider a more neutral sentence: “…visualized in FigTree 1.4.0 and finalized in Adobe Illustrator CS 2015.”

corrected

Line 175.177: Although the meaning of the sentence is clar, I believe it can be improved for better readability (e.g.: Since our goal is to clarify the taxonomy of Flabellinidae s.l. at the genus and family level, species diversity and identity were not the focus, and no species delimitation analyses were performed.”)

corrected

Line 178–179: Very long; suggest breaking into two sentences for readability.

corrected

Line 183: Consider being more concise: “…examined for external morphology under a stereomicroscope.”

corrected

Line 193–194: Could be shortened to: “…dissected dorsally along the midline and examined under a stereomicroscope.”

corrected

Results – Molecular Phylogeny

General comments: Without dismissing the quality of the research (that is of high standards) the section is overall, very dense and can overwhelm the reader.

corrected

• Please, consider summarizing some repetitive descriptions (e.g., node supports, repeated morphological states) and referring readers to supplementary tables/figures. (e.g., lines 212 to 225)

corrected

• The text describes complex clades in a single long sentence in several parts of the text. Breaking these into shorter sentences would improve readability.

corrected

• Figures are clear, but cross-referencing between text and figures could be more direct (e.g., “see Fig. 1A” rather than just “see Fig. 1”) and included in more parts of the results.

Fig. 1 does not have any parts (A, B, etc), but we revised all links to figures to ensure that all parts were cited where applicable

• Please ensure consistent use of family and genus names (e.g., Flabellinidae s.str. vs. Flabellinidae sensu stricto).

We checked this and did not find any use of ‘sensu lato’ or ‘sensu stricto’

In some places, terms like “recovered” and “clustered” could be simplified to avoid repetition.

corrected

• The morphological analysis. Consider highlighting which traits were most informative for higherrank taxonomy, and which ones were too variable to be useful.

All mentioned traits are useful for high-rank taxonomy, but depending on the group, some traits may have a higher importance. Listing this is difficult, as the diversity is very high. The figure 2 perfectly shows what traits are variable within each taxon, and what are taxon-specific.

• Some detailed species-level results (e.g., multiple clades of Samla bicolor) could be shortened. This would keep the focus on the main findings relevant to higher-level taxonomy.

Done!

Line 197: “from 130” instead of “for”

corrected

Line 199: Change the position of latter: “the latter two partitions were trimmed with Gblocks.”

corrected

Line 203–204: The use of “compromise” sounds weird, consider changing “…and of the genera

Edmundsella Korshunova et al., 2017 and Samla was compromised…” for something as “…and the genera Edmundsella and Samla were not recovered as monophyletic…”

corrected

Line 217: “…split into two distinct sister clades…” is redundant, I´d say “…split into two distinct clades…” instead.

corrected

Line 222–224: Modify: “unexpectedly grouped in a monophyletic clade…”

corrected

Line 236–238: Consider replacing “…which conforms previously recovered relationships…” by “…which are consistent with previously recovered relationships…”

corrected

Line 240: Can be simplify as “The ML tree suggested a different topology among subclades, but without support.”

corrected

Line 246: Correct as “…was recovered as sister to Coryphella…”

corrected

Line 247: “Add comma: “Within Flabellinidae s.str., three large clades were recovered.”

corrected

Line 265–269: These sentences are a bit difficult to follow. Please rephrase.

corrected

Line 282–284: Consider rephrasing, e.g “…but this relationship lacked support in both BI (PP = 0.74) and ML (BS = 48).”

corrected

Line 291–293: Add comma: “With the exclusion of S. rubropurpurata and S. telja, the genus Samla was recovered as monophyletic…”

corrected

Line 299–301: Simplify “…including Piseinotecus soussi…”

corrected

Line 305–306: Modify “…instead it formed a polytomy with the other P. goddardi specimens and P. amica…”

corrected

Line 313–315: I´d be more cautious with the sentence maybe something like “…suggesting they may represent two distinct species.”

corrected

Results – Systematics

The Systematics section is well structured and supported by valuable details. It shows clear expertise, and the supplementary tables compiling all morphological characters are particularly valuable for future research. The descriptions are generally clear and informative. However, several family- and genus-level diagnoses are too broad, relying mainly on variable traits. Refining these diagnoses to emphasize consistent or unique character combinations would make them more useful and would provide stronger morphological support to complement the molecular justification for the new genera.

We think that expanded diagnoses could be more consistent, if they will identify comparable character states across different taxa. As Flabellinidae morphological diversity is extremely high and several taxa still require a comprehensive revision, broader diagnoses are better than shortened.

Discussion

General comments: As in previous sections, the discussion is through and demonstrates strong knowledge. This section effectively places the results in the context of previous works and highlights the challenges of these groups and the highly morphological variability. The section is, again, a bit dense with long sentences that sometimes obscure the key messages. I am adding here some suggestions, but the text should be revised to make it more straightforward (e.g., avoid repetitions, short sentences, avoid repeating results).

corrected

Line 1274: I suggest reducing the sentence “…is polyphyletic, corroborating the revision by [22].” (more concise).

corrected

Line 1276- 1279: I suggest simplifying the sentence: “Representatives of the families Apataidae, Samlidae and Piseiontecidae form three highly supported and derived clades, not related to the ‘core’ group of Flabellinidae nor to each other (Fig. 1).”

corrected

Can you compare this with previous studies? If so, add reference.

added

Line 1279–1284: Rephrase to avoid repetitive language and simplify (this is already explained above) “We tentatively suggest synonymizing Unidentiidae and Piseinotecidae, but this should be reconsidered once molecular data for the type species P. divae become available”.

corrected

Line 1286–1292: The sentence listing synapomorphies is long. Break into two sentences or eliminate redundant information as “of Samlidae lines 1289-1290).

Add references (I understand that have been already discussed refers to previous works).

corrected

I suggest rephrasing and simplifying “In the case of Samlidae we believe that”(Line 1291) to: “we consider that these synapomorphies have diagnostic value”.

corrected

I consider that “and should facilitate the correct placement of newly described taxa even if molecular data” is redundant, that is the meaning of diagnostic value.

corrected

Line 1295–1297: I suggest emphasizing the unexpected nature of the relationship “…a possible sister relationship between Samlidae and Notaeolidiidae, an unexpected result given their markedly different morphologies.”

corrected

Also change the order in which you are presenting the characters so you start with Samlidae.

corrected

No need to use connectors as “At the same time” to simplify sentences.

corrected

Line 1304–1307: Could be shorter as for example “…Contrary to previous studies, the expected sister relationship of Cumanotidae and Pseudovermidae was not recovered.”

corrected

Line 1311 -1314:Rephrase to improve readability “The resulted relationships (Notaeolidiidae + Samlidae; Cumanotidae + Apataidae) should be considered as tentative, since our analysis lacks many aeolid taxa, including lineages of the large family Facelinidae s.l.”

corrected

Line 1314: Eliminate “We also believe” from the sentence.

corrected

Line 1320-1323: This can be reduced by eliminating part of the sentence as you are already referring to the figure “At the same time, Moreover in all our phylogenetic analyses Bajaeolis is recovered in a clade including representatives of the family Flabel

---

## [Decision Letter · Decision Letter 1]

7 Apr 2026

Neither “lumpers” nor “splitters”: A global revision of Flabellinidae s.l. nudibranchs (Gastropoda: Heterobranchia: Nudibranchia)

PONE-D-25-43242R1

Dear Dr. Ekimova,

We’re pleased to inform you that your manuscript has been judged scientifically suitable for publication and will be formally accepted for publication once it meets all outstanding technical requirements.

An invoice will be generated when your article is formally accepted. Please note, if your institution has a publishing partnership with PLOS and your article meets the relevant criteria, all or part of your publication costs will be covered. Please make sure your user information is up-to-date by logging into Editorial Manager at Editorial Manager® and clicking the ‘Update My Information' link at the top of the page. For questions related to billing, please contact  and clicking the ‘Update My Information' link at the top of the page. For questions related to billing, please contact billing support..

Kind regards,

Michael Schubert

Academic Editor

PLOS One

Reviewers' comments:

Reviewer's Responses to Questions

**Comments to the Author**

1. If the authors have adequately addressed your comments raised in a previous round of review and you feel that this manuscript is now acceptable for publication, you may indicate that here to bypass the “Comments to the Author” section, enter your conflict of interest statement in the “Confidential to Editor” section, and submit your "Accept" recommendation.

Reviewer #1: All comments have been addressed

Reviewer #3: All comments have been addressed

2. Is the manuscript technically sound, and do the data support the conclusions?

Reviewer #1: Yes

Reviewer #3: Yes

3. Has the statistical analysis been performed appropriately and rigorously? 

Reviewer #1: N/A

Reviewer #3: Yes

4. Have the authors made all data underlying the findings in their manuscript fully available?

Reviewer #1: Yes

Reviewer #3: Yes

5. Is the manuscript presented in an intelligible fashion and written in standard English?

Reviewer #1: Yes

Reviewer #3: Yes

6. Review Comments to the Author

Reviewer #1: The authors have satisfactorily addressed most of the previous comments, and the manuscript has clearly improved in terms of clarity. The study represents a substantial and valuable contribution supported by a large dataset and a rigorous integrative approach.

I understand and appreciate the authors’ rationale for maintaining relatively broad diagnoses given the high morphological variability of the group. However, I still believe that a slight reduction (focusing on the most consistent and informative character combinations) would improve their practical utility and readability without compromising detail. Similarly, although the abstract has been revised, it remains somewhat dense to me. A further reduction, emphasizing the main findings, would enhance its impact.

Nevertheless, these are relatively minor points. Overall the manuscript is well executed, provides relevant findings and thus I consider is suitable for publication.

Reviewer #3: (No Response)

7. PLOS authors have the option to publish the peer review history of their article (what does this mean?). If published, this will include your full peer review and any attached files.). If published, this will include your full peer review and any attached files.

.

Reviewer #1: No

Reviewer #3: **Yes:** Jessica GoodheartJessica Goodheart

---

## [Editor Report · Acceptance letter]

PONE-D-25-43242R1

PLOS One

Dear Dr. Ekimova,

I'm pleased to inform you that your manuscript has been deemed suitable for publication in PLOS One. Congratulations! Your manuscript is now being handed over to our production team.

Kind regards,

on behalf of

Dr. Michael Schubert

Academic Editor

PLOS One